# Intrinsic ecological dynamics drive biodiversity turnover in model metacommunities

Jacob D. O'Sullivan [1✉], J. Christopher D. Terry [1] & Axel G. Rossberg[1]

Turnover of species composition through time is frequently observed in ecosystems. It is often interpreted as indicating the impact of changes in the environment. Continuous turnover due solely to ecological dynamics—species interactions and dispersal—is also known to be theoretically possible; however the prevalence of such autonomous turnover in natural communities remains unclear. Here we demonstrate that observed patterns of compositional turnover and other important macroecological phenomena can be reproduced in large spatially explicit model ecosystems, without external forcing such as environmental change or the invasion of new species into the model. We find that autonomous turnover is triggered by the onset of ecological structural instability—the mechanism that also limits local biodiversity. These results imply that the potential role of autonomous turnover as a widespread and important natural process is underappreciated, challenging assumptions implicit in many observation and management tools. Quantifying the baseline level of compositional change would greatly improve ecological status assessments.

[1] School of Biological and Chemical Sciences, Queen Mary University of London, London, UK. ✉email: j.osullivan@qmul.ac.uk

Change in species composition observed in a single location through time, called community turnover, is observed in all natural ecosystems. Potential drivers of such biotic change include changes in the abiotic environment, random demographic fluctuations (referred to as community drift) and population dynamics driven by ecological interactions and dispersal. Analysis of community time series suggests communities turn over at a faster rate than can be explained by random drift[1,2]. Climate change and other anthropogenic pressures are known to contribute to community turnover[3–7] and there is evidence to suggest that turnover is accelerating in some biomes[8].

Ecological assessments, projections and mitigation strategies are therefore commonly designed around the assumption that communities turn over predominantly in response environmental change and direct anthropogenic pressures[9]. The extent to which processes intrinsic to ecosystems contribute to turnover, however, remains poorly understood[10]. Understanding the expected amount of temporal turnover due to such intrinsic processes is of vital importance if ecological change is to be accurately interpreted[7]. If strong temporal community turnover were a natural phenomenon that can arise independently of changes in the abiotic environment, then observed shifts in the composition of ecological communities would not on their own demonstrate external pressures.

In theoretical models of ecological communities population abundances do not necessarily arrive at fixed points. Instead, such systems can manifest persistent dynamics which we refer to here as 'autonomous' since they do not depend on variation in the external environment or other extrinsic drivers. When these population fluctuations are strong, changes in the abundances of species can be dramatic and even drive species locally extinct; if an excluded species retains occupancy in adjacent patches[11], it may re-colonise at some future time. We refer to as 'autonomous turnover' local compositional changes involving colonisation-extinction processes or significant restructuring of relative abundances, driven by such autonomous population dynamics.

Limited availability of historical turnover data before the onset of widespread anthropogenic impacts poses considerable challenges when trying to establish the natural baseline of turnover. Nevertheless, broad consistency amongst the species-time-area relationships observed in extant assemblages[12,13] points to a consistency in the dominant underlying biological process. It is reasonable to expect, therefore, that the drivers of such spatio-temporal turnover can be probed using theoretical models.

Previous theoretical[11,14–17] and experimental studies[18] have shown how competitive ecological communities (for specific network structures or parameter combinations) can generate any type of dynamical behaviour, including persistent chaotic cycles. Likewise, antagonistic interactions between predator and prey species have been shown in both theory and experiment to lead to persistent population oscillations in the absence of external variation[19,20]. However, these cyclic processes are different from, and have not usually been associated with, empirical observations of acyclic, directional compositional turnover[1,2]. An important distinction between these processes lies in the role of space. While cyclic forms of community dynamics can lead to characteristic spatial structures[17,18,21], cyclic dynamics do not in principle require space[19,20]. Acyclic turnover, on the other hand, manifestly involves colonisation by species from surrounding patches and therefore explicitly requires that a community is embedded in a spatially structured ecological neighbourhood.

Here we ask: can community dynamics enabled by spatial structure account for the observed macroecological patterns in compositional turnover? We address this question drawing on recent advances in the theory of spatially extended ecological communities, so called "metacommunities"[22], using a population-dynamical simulation model with explicit spatial and environmental structure[23] that has been shown to reproduce fundamental *spatial* biodiversity patterns. As previously shown, such metacommunity models can exhibit a phenomenon called "ecological structural instability"[24], as a result of which species richness at both local and regional scales is intrinsically regulated[23]. The structural stability of a system refers to its capacity to sustain changes in parameters without undergoing qualitative changes in dynamical behaviour[25]. As such, ecological structural stability is taken to describe in particular the capacity of a community to *persist* in the face of small biotic or abiotic perturbations[24,26–30]. Empirical observation of many of the emergent phenomena associated with ecological structural instability provides compelling indirect evidence for the prevalence of structural instability in nature[23,31]. Our understanding of the impact of structurally unstable diversity regulation on temporal community-level properties, however, remains incomplete[32]. Here we build upon earlier work by exploring the spatio-*temporal* patterns that emerge in metacommunity models. We find that, when expanding the spatial and taxonomic scale of simulations beyond those studied previously, metacommunities manifest autonomous compositional turnover which can be substantial and, in the case of large models, acyclic. This turnover is best understood as continuous fluctuations of community state around heteroclinic networks characterised by multiple unstable equilibria. Crucially, the macroecology of these heteroclinic networks matches known empirical patterns.

## Results and discussion

**Metacommunity model and asymptotic community assembly.** We built a large set of model metacommunities (detailed in full in "Methods") describing competitive dynamics within a single guild of species across a landscape. Each metacommunity consisted of a set of patches, or local communities, randomly placed in a square arena and linked by a spatial network. The dynamics of each population are governed by three processes: inter- and intraspecific interactions, heterogeneous responses to the environment and dispersal between adjacent patches (Fig. 1). Competition coefficients between species are drawn at random and the population dynamics within each patch are described by a Lotka-Volterra competition model. We control the level of environmental heterogeneity across the network directly by generating an intrinsic growth rate for each species at each patch from a random, spatially correlated distribution. To ensure any turnover is purely autonomous, we keep the environment fixed throughout simulations. Dispersal between neighbouring patches declines exponentially with distance between sites. This formulation allows precise and independent control of key properties of the metacommunity–the number of patches, the characteristic dispersal length and the heterogeneity of the environment.

To populate the model metacommunities, we iteratively introduced species with randomly generated intrinsic growth rates and interspecific interaction coefficients. Between successive regional invasions we simulated the model dynamics, and removed any species whose abundance fell below a threshold across the whole network. Through this assembly process and the eventual onset of ecological structural instability, both average local diversity, the number of species coexisting in a given patch, and regional diversity, the total number of species in the metacommunity, eventually saturate and then fluctuate around an equilibrium value—any introduction of a new species then leads on average to the extinction of one other species (Supplementary Fig. 1). In these intrinsically regulated metacommunities we then studied the phenomenology of autonomous community turnover *in the absence of regional invasions or abiotic change*.

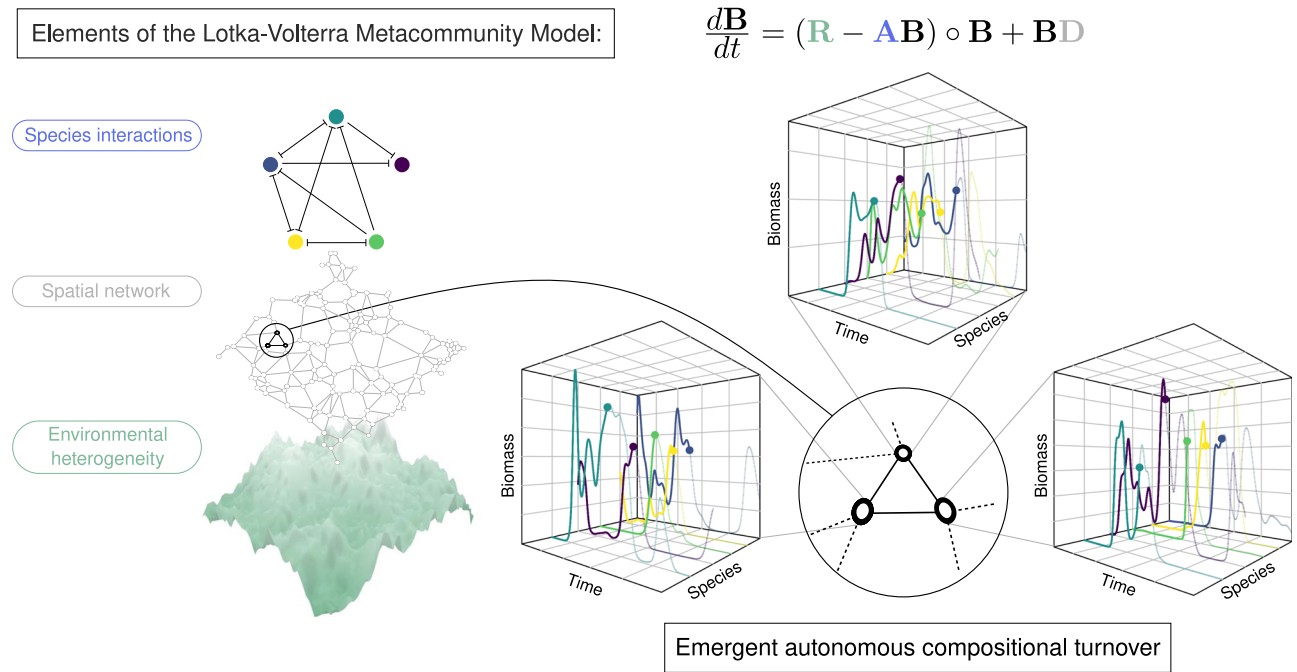

$$\frac{d\mathbf{B}}{dt} = (\mathbf{R} - \mathbf{A}\mathbf{B}) \circ \mathbf{B} + \mathbf{B}\mathbf{D}$$

**Fig. 1 Elements of the Lotka-Volterra metacommunity model and the emergence of autonomous population dynamics.** Environmental heterogeneity, represented by the intrinsic growth rate matrix **R**, is modelled using a spatially autocorrelated Gaussian random field. A random spatial network, represented by the dispersal matrix **D**, defines the spatial connectivity of the landscape. The network of species interactions, represented by the competitive overlap matrix **A**, is modelled by sampling competition coefficients at random (perpendicular bars indicate recipients of a deleterious competitive impact). The resulting dynamics of local population biomasses, given by the colour-coded equation, are numerically simulated. The Hadamard product '∘' represents element-wise matrix multiplication. For large metacommunities, local populations exhibit persistent dynamics despite the absence of external drivers. In the 3D boxes, typical simulated biomass dynamics of dominating species are plotted on linear axes over 2500 unit times. The graphs illustrate the complexity of the autonomous dynamics and the propensity for compositional change (local extinction and colonisation).

In our metacommunity model, local community dynamics and therefore local limits on species richness depend on a combination of biotic and abiotic filtering (non-uniform responses of species to local conditions)[33–35] and immigration from adjacent patches, generating so called mass effects in the local community[36–38]. Biotic filtering via interspecific competition is encoded in the interaction coefficients $A_{ij}$, while abiotic filtering occurs via the spatial variation of intrinsic growth rates $R_{ix}$. For simplicity, and since predator-prey dynamics are known to generate oscillations[39] through mechanisms distinct from those we report here, we restrict our analysis to competitive communities for which all ecological interactions are antagonistic. The off-diagonal elements of the interaction matrix **A** describe how one species $i$ affects another species $j$. These are sampled independently from a discrete distribution, such that the interaction strength $A_{ij}$ is set to a constant value in the range 0 to 1 (in most cases 0.5) with fixed probability (connectance, in most cases 0.5) and otherwise set to zero. Intraspecific competition coefficients $A_{ii}$ are set to 1 for all species. This discrete distribution of the interaction terms was chosen for its relative efficiency. In the Supplementary discussion (and Supplementary Fig. 2) we show that outcomes remain unaffected when more complex distributions are modelled. Intrinsic growth rates $R_{ix}$ are sampled from spatially correlated normal distributions with mean 1, autocorrelation length $\phi$ and variance $\sigma^2$ (Supplementary Fig. 3).

Dispersal is modelled via a spatial connectivity matrix with elements $D_{xy}$. The topology of the model metacommunity, expressed through **D**, is generated by sampling the spatial coordinates of $N$ patches from a uniform distribution $\mathcal{U}(0, \sqrt{N}) \times \mathcal{U}(0, \sqrt{N})$, i.e., an area of size $N$. Thus, under variation of the number of patches, the inter-patch distances remain fixed on average. Spatial connectivity is defined by linking these patches through a Gabriel graph[40], a planar graph generated by an algorithm that, on average, links each local community to four close neighbours[41]. Avoidance of direct long-distance dispersal and the sparsity of the resulting dispersal matrix permit the use of efficient numerical methods. The exponential dispersal kernel defining $D_{xy}$ is tuned by the dispersal length $\ell$, which is fixed for all species.

The dynamics of local population biomasses $B_{ix} = B_{ix}(t)$ are modelled using a system of spatially coupled Lotka-Volterra (LV) equations that, in matrix notation, takes the form[23]

$$\frac{d\mathbf{B}}{dt} = \mathbf{B} \circ (\mathbf{R} - \mathbf{A}\mathbf{B}) + \mathbf{B}\mathbf{D}, \qquad (1)$$

with ∘ denoting element-wise multiplication. Hereafter this formalism is referred to as the Lotka-Volterra Metacommunity Model (LVMCM). Further technical details are provided in Methods and the Supplementary Discussion.

In order to numerically probe the impacts of $\ell$, $\phi$ and $\sigma^2$ on the emergent temporal dynamics, we initially fixed $N = 64$ and varied each parameter through multiple orders of magnitude (Supplementary Fig. 4). In order to obtain a full characterisation of autonomous turnover in the computationally accessible spatial range ($N \le 256$), we then selected a parameter combination found to generate substantial fluctuations for further analysis. Thereafter we assembled metacommunities of 8–256 patches (Fig. 2a) until regional diversity limits were reached (with tenfold replication) and generated community time series of $10^4$ unit times from which the phenomenology of autonomous turnover could be explored in detail. We found no evidence to suggest that the phenomenology described below depends on this specific parameter combination. While future results may confirm or

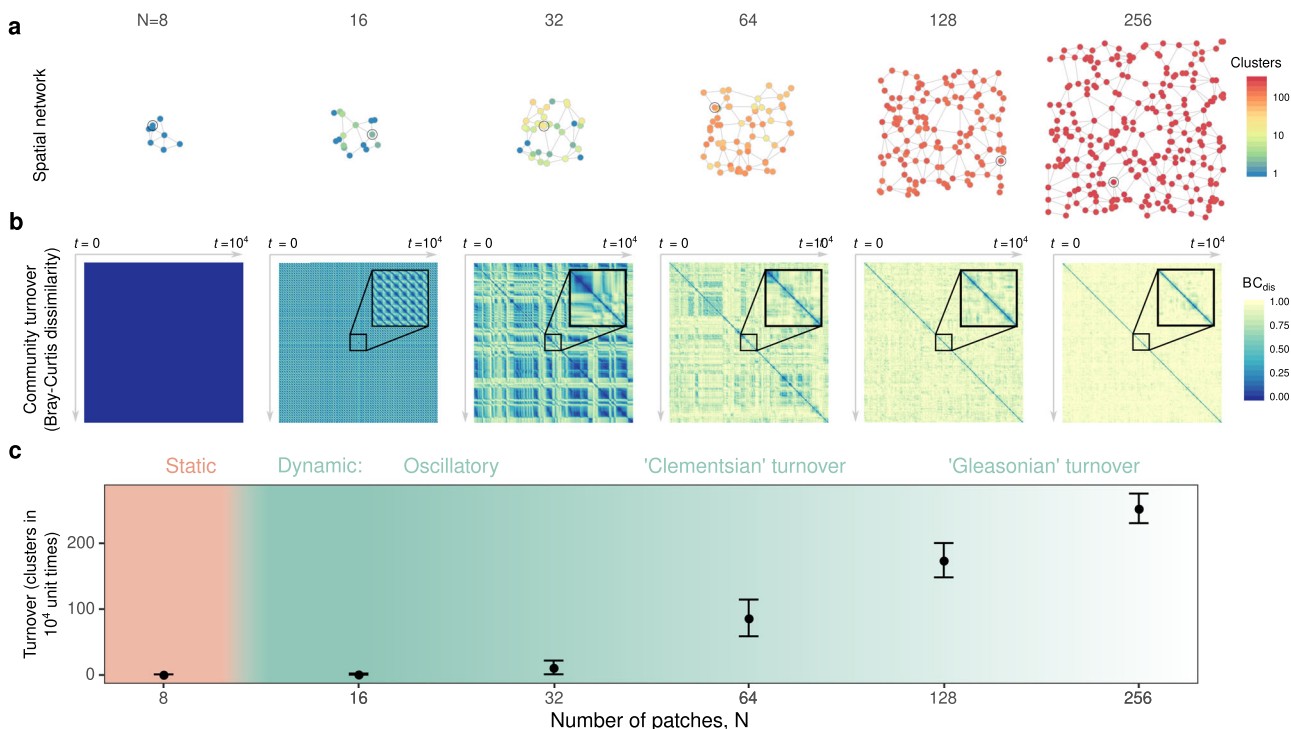

**Fig. 2 Autonomous turnover in model metacommunities. a** Typical model metacommunities: a spatial network with $N$ nodes representing local communities (or patches) and edges, channels of dispersal. Patch colour represents the number of clusters in local community state space detected over $10^4$ unit times $t$ using hierarchical clustering of the Bray-Curtis (BC) dissimilarity matrix, Supplementary Fig. 6. **b** Colour coded matrices of pairwise temporal BC dissimilarity corresponding to the circled patches in (**a**). Insets represent $10^2$ unit times. For small networks ($N = 8$) local compositions converge to static fixed points. As metacommunity extent increases, however, persistent dynamics emerge. Initially this autonomous turnover is oscillatory in nature with communities fluctuating between small numbers of states which can be grouped into clusters ($16 \leq N \leq 32$). Intermediate metacommunities ($32 \leq N \leq 64$) manifest "Clementsian" temporal turnover, characterised by sharp transitions in composition, implying species turn over in cohorts. Large metacommunities ($N \geq 128$) turn over continuously, implying "Gleasonian" assembly dynamics in which species' temporal occupancies are independent. **c** The mean number of local compositional clusters detected for metacommunities of various numbers of patches $N$ (error bars represent standard deviation across all replicated simulations). While the transition from static to dynamic community composition at the local scale is sharp (see text), non-uniform turnover *within* metacommunities (**a**) blurs the transition at the regional scale. $A_{ij} = 0.5$ with probability 0.5, $\phi = 10$, $\sigma^2 = 0.01$, $\ell = 0.5$.

refute this, autonomous turnover arises over a wide range of parameters (Supplementary Fig. 4) and as such the phenomenon is robust.

**Autonomous turnover in model metacommunities.** For small ($N \leq 8$) metacommunities assembled to regional diversity limits, populations attain equilibria, i.e., converge to fixed points, implying the absence of autonomous turnover[23]. With increasing metacommunity size $N$, however, we observe the emergence of persistent population dynamics (Supplementary Fig. 5 and external video) that can produce substantial turnover in local community composition. This autonomous turnover can be represented through Bray-Curtis[42] (BC) dissimilarity matrices comparing local community composition through time (Fig. 2b), and quantified by the number of compositional clusters detected in such matrices using hierarchical cluster analysis (Fig. 2a, c).

At intermediate spatial scales (Fig. 2, $16 \leq N \leq 32$) we often find oscillatory dynamics, which can be perfectly periodic or slightly irregular. With increasing oscillation amplitude, these lead to persistent turnover dynamics where local communities repeatedly transition between a small number of distinct compositional clusters (represented in Fig. 2 by stripes of high pairwise BC dissimilarity spanning large temporal ranges). At even larger scales ($N \geq 64$) this compositional coherence begins to break down, and for very large metacommunities ($N \geq 128$) autonomous dynamics drive continuous acyclic change in community composition. The number of compositional clusters detected over time typically varies within a given metacommunity (Fig. 2a node colour), however we find a clear increase in the average number of compositional clusters, i.e., an increase in turnover, with increasing total metacommunity size (Fig. 2c).

Metacommunities in which the boundaries of species ranges along environmental gradients are clumped are termed *Clementsian*, while those for which range limits are independently distributed are referred to as *Gleasonian*[43]. We consider the block structure of the temporal dissimilarity matrix at intermediate $N$ to represent a form of Clementsian temporal turnover, characterised by sudden significant shifts in community composition. Metacommunity models similar to ours have been found to generate such patterns along spatial gradients[44], potentially via an analogous mechanism[45]. Large, diverse model metacommunities manifest Gleasonian temporal turnover. In such cases, species colonisations and extirpations are largely independent and temporal occupancies predominantly uncorrelated, such that compositional change is continuous, rarely, if ever, reverting to the same state.

**Mechanistic explanation of autonomous turnover.** Surprisingly, the onset and increasing complexity of autonomous turnover as system size $N$ increases (Fig. 2) can be understood as a consequence of local community dynamics alone. To explain this, we first recall relevant theoretical results for isolated LV communities. Then we demonstrate that, in presence of weak propagule pressure, these results imply local community turnover dynamics,

controlled by the richness of potential invaders, that closely mirror the dependence on system size seen in full LV metacommunities.

Application of methods from statistical mechanics to models of large isolated LV communities with random interactions revealed that such models exhibit qualitatively distinct phases[46–48]. If the number of modelled species, $S$, interpreted as species pool size, lies below some threshold value determined by the distribution of interaction strengths (Supplementary Fig. 7), these models exhibit a unique linearly stable equilibrium (Unique Fixed Point phase, UFP). Some species may go extinct, but the majority persists[48]. When pool size $S$ exceeds this threshold, there appear to be no more linearly stable equilibrium configurations. Any community formed by a selection from the $S$ species is either unfeasible (there is no equilibrium with all species present), intrinsically linearly unstable, or invadable by at least one of the excluded species. This has been called the multiple attractor (MA) phase[47]. However, the implied notion that this part of the phase space is in fact characterised by multiple stable equilibria may be incorrect.

Population dynamical models with many species have been shown to easily exhibit attractors called stable heteroclinic networks[49], which are characterised by dynamics in which the system bounces around between several unstable equilibria, each corresponding to a different composition of the extant community, implying indefinite, autonomous community turnover (Fig. 3, red line). As these attractors are approached, models exhibit increasingly long intermittent phases of slow dynamics, which, when numerically simulated, can give the impression that the system eventually reaches one of several 'stable' equilibria, suggesting that turnover comes to a halt. We demonstrate in the Supplementary discussion that the MA phase of isolated LV models is in fact characterised by such stable heteroclinic networks (Supplementary Figs. 8 and 9). Note, we retain the MA terminology here because the underlying complete

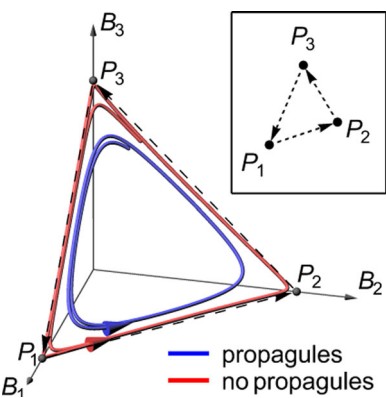

**Fig. 3 Approximate heteroclinic networks underlie autonomous community turnover.** The main panel shows two trajectories in the state space of a community of three hypothetical species (population biomasses $B_1$, $B_2$, $B_3$) that are in non-hierarchical competition with each other, such that no species can competitively exclude both others (a "rock-paper-scissors game"[17]). Without propagule pressure, the system has three unstable equilibrium points ($P_1$, $P_2$, $P_3$) and cycles between these (red curve), coming increasingly close to the equilibria and spending ever more time in the vicinity of each. The corresponding attractor is called a *heteroclinic cycle* (dashed arrows). Under weak extrinsic propagule pressure (blue curve), the three equilibria and the heteroclinic cycle disappear, yet the system closely tracks the original cycle in state space. Such a cycle can be represented as a graph linking the dynamically connected equilibria (inset). With more interacting species, these graphs can become complex "heteroclinic networks"[49–51] with trajectories representing complex sequences of species composition during autonomous community turnover.

heteroclinic networks, interpreted as a directed graph[50,51] (Fig. 3, inset), might have multiple components that are mutually unreachable through dynamic transitions[52], each representing a different attractor.

If one now adds to such isolated LV models terms representing weak propagule pressure for all $S$ species (Supplementary Eq. (2)), dynamically equivalent to mass effects occurring in the full metacommunity model (Eq. (1)), then none of the $S$ species can entirely go extinct. The weak influx of biomass drives community states away from the unstable equilibria representing coexistence of subsets of the $S$ species and the heteroclinic network connecting them (blue line in Fig. 3). Typically, system dynamics then still follow trajectories closely tracking the original heteroclinic networks (Fig. 3), but now without requiring boundless time to transition from the vicinity of one equilibrium to the next.

The nature and complexity of the resulting population dynamics depend on the size and complexity of the underlying heteroclinic network, and both increase with pool size $S$. In simulations (Supplementary Fig. 10) we find that, as $S$ increases, LV models with weak propagule pressure pass through the same sequence of states as we documented for LVMCM metacommunities in Fig. 2: equilibria, oscillatory population dynamics, Clementsian and finally Gleasonian temporal turnover.

Above we introduced the number of clusters detected in Bray-Curtis dissimilarity matrices of fixed time series length as a means of quantifying the approximate number of equilibria visited during local community turnover. As shown in Fig. 4a, b, this number increases in LV models with $S$ in a manner strikingly similar to its increase in the LVMCM with the number of species present in the ecological neighbourhood of a given patch. Thus, dynamics within a patch are controlled not by $N$ directly but rather by neighbourhood species richness. For a given neighbourhood, species richness depends on the number of connected patches, the total area and therefore total abiotic heterogeneity encompassed, and the connectivity, all of which can vary substantially within a metacommunity of a given size $N$. As illustrated in Fig. 4b, there is a tendency for neighbourhood richness to be larger in larger metacommunities, leading indirectly to the dependence of metacommunity dynamics on $N$ seen in Fig. 2.

There is thus a close correspondence between dynamically isolated LV models and LVMCM metacommunities, in the sequence of dynamic states as propagule richness increases and in the resulting complexity of dynamics quantified by counting compositional clusters. This suggests that underlying heteroclinic networks, which are revealed by adding propagule pressure in isolated communities, explain the complex dynamics seen in LVMCM metacommunities.

For the isolated LV community, the threshold beyond which autonomous turnover is detected (>1 compositional cluster) occurs at a pools size of around $S = 35$ species, consistent with the theoretical prediction[47] of the transition between the UFP and MA phases (Supplementary Discussion). Close inspection of this threshold reveals an important and hitherto unreported relationship between the transition into the MA phase and local ecological limits set by the onset of ecological structural instability, which is known to regulate species richness in LV systems subject to external invasion pressure[23,24]: in the Supplementary Discussion we show that the boundary between the UFP and MA phases[47] coincides precisely with the onset of structural instability[24] (Supplementary Eqs. (3)–(9)).

For LVMCM metacommunities, this relationship (demonstrated analytically in the Supplementary Discussion) is numerically confirmed in Fig. 5. During assembly, local species richness increases until it reaches the limit imposed by local structural

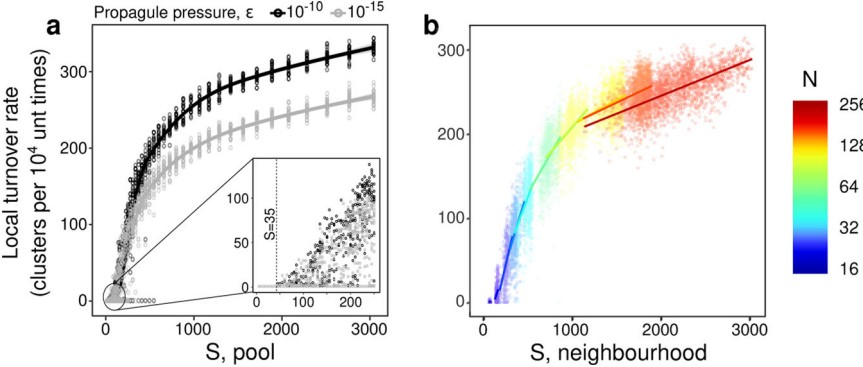

**Fig. 4 Ecological mass effects drive autonomous turnover. a** The number of compositional clusters detected, plotted against the size of the pool of potential invaders $S$ for an isolated LV community using a propagule pressure $\epsilon$ of $10^{-10}$ and $10^{-15}$, fit by a generalised additive model[87]. For $S < 35$ a single cluster is detected. For $S \geq 35$ autonomous turnover occurs ($\geq 1$ compositional clusters) with the transition indicated by the dashed line (inset). **b** Qualitatively identical behaviour was observed for model metacommunities in which "propagule pressure" arises due to ecological mass effects from the local neighbourhood. Each point represents a single patch. Lines in (**b**) are standard linear regressions. The good alignment of subsequent fits demonstrates that neighbourhood diversity is the dominating predictor of cluster number, rather than patch number $N$. $A_{ij} = 0.5$ with probability 0.5, $\phi = 10$, $\sigma^2 = 0.01$, $\ell = 0.5$.

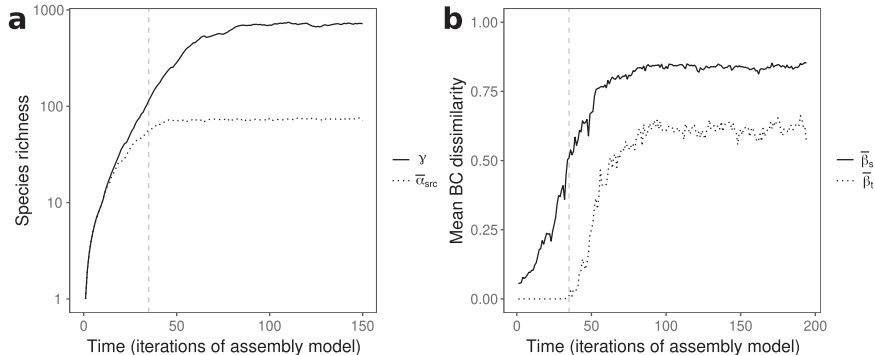

**Fig. 5 The emergence of temporal turnover during metacommunity assembly. a** Local species richness, defined by reference to source populations only ($\overline{\alpha}_{src}$, grey) and regional diversity ($\gamma$ black) for a single metacommunity of $N = 32$ coupled communities during iterative regional invasion of random species. We quantify local source diversity $\overline{\alpha}_{src}$ as the metacommunity average of the number $\alpha_{src}$ of non-zero equilibrium populations persisting when immigration is switched off (off-diagonal elements of **D** set to zero), since this is the component of a local community subject to strict ecological limits to biodiversity. Note the log scale chosen for easy comparison of local and regional species richness. **b** Increases in regional diversity beyond local limits arise via corresponding increases in spatial turnover ($\overline{\beta}_s$, black). Autonomous temporal turnover ($\overline{\beta}_t$, grey) sets in (crosses a threshold mean Bray Curtis (BC) dissimilarity of $10^{-2}$) precisely when average local species richness $\overline{\alpha}_{src}$ has reached its limit, reflecting the equivalence of the transition to the MA phase space and the onset of local structural instability. In both panels, the dashed line marks the point at which autonomous temporal turnover was first detected. $A_{ij} = 0.3$ with probability 0.3, $\phi = 10$, $\sigma^2 = 0.01$, $\ell = 0.5$. Both spatial and temporal turnover computed as the mean BC dissimilarity. In each iteration of the assembly model (regional invasion event), $0.1S + 1$ species were introduced. Dynamics were simulated for $2 \times 10^4$ unit times, with the second $10^4$ unit times analysed for autonomous turnover, and a total of $10^4$ invasions were modelled.

instability. Further assembly occurs via the "regionalisation" of the biota[53]—a collapse in average range sizes[23] and associated increase in spatial beta diversity—until regional diversity limits are reached[23]. The emergence of autonomous turnover coincides with the onset of species saturation at the local scale. Autonomous turnover can therefore serve as an indirect indication of intrinsic biodiversity regulation via local structural instability in complex communities.

Thus, we have shown that propagule pressure perturbs local communities away from unstable equilibria and drives compositional change. In order to invade, however, species need to be capable of passing through biotic and abiotic filters[33–35]. We would expect, therefore, that turnover would be suppressed in highly heterogeneous or poorly connected environments where mass effects are weak. Indeed, by manipulating the autocorrelation length $\phi$ and variance $\sigma^2$ of the abiotic filter represented by the matrix **R** and the characteristic dispersal length $\ell$, we observe a sharp drop-off in temporal turnover in parameter regimes that

maximise between-patch community dissimilarity (short environmental correlation or dispersal lengths, Supplementary Fig. 11). Thus, we conclude that it is not species richness or spatial dissimilarity per se that best predict temporal turnover, but the size of the pool of species with positive invasion fitness, i.e., those not repelled by the combined effects of biotic and abiotic filters.

**The macroecology of autonomous turnover.** We find good correspondence between temporal and spatio-temporal biodiversity patterns emerging in model metacommunities in the absence of external abiotic change and in empirical data (Fig. 6), with quantitative characteristics lying within the ranges observed in natural ecosystems.

*Temporal occupancy.* The proportion of time in which species occupy a community tends to have a bi-modal empirical distribution[54–56] (Fig. 6a). The distribution we found in simulations (Fig. 6e) closely matches the empirical pattern.

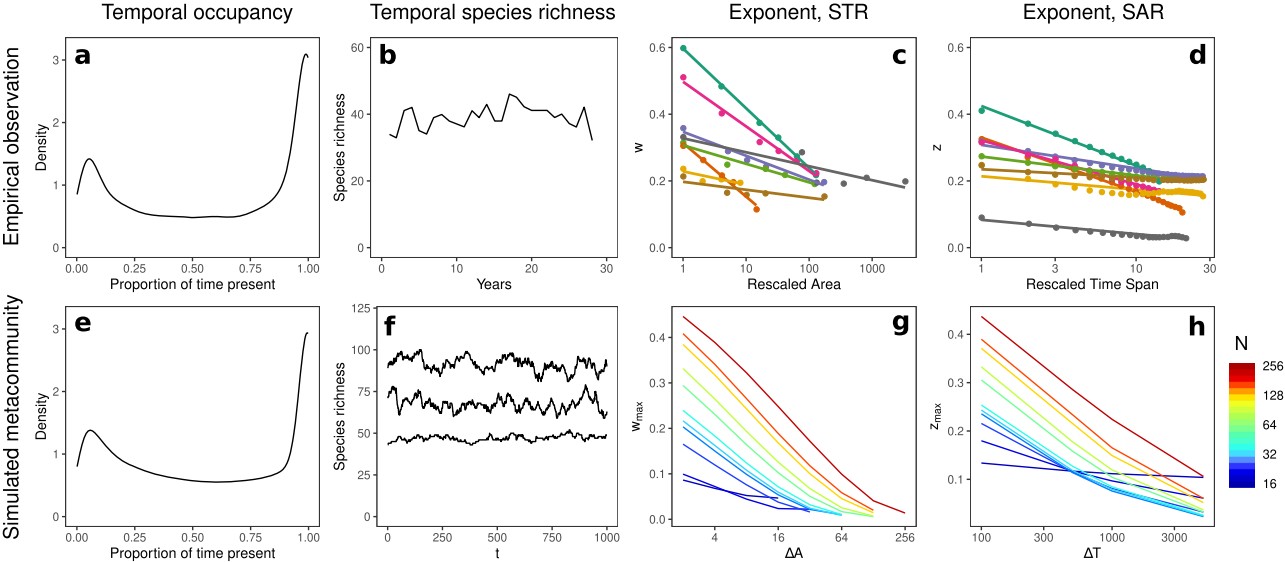

**Fig. 6 Macroecological signatures of autonomous compositional change.** A bimodal distribution in temporal occupancy observed in North American birds[54] (**a**) and in simulations (**e** $N = 64$, $\phi = 5$, $\sigma^2 = 0.01$, $\ell = 0.5$). Intrisically regulated species richness observed in estuarine fish species[59] (**b**) and in simulations (**f** $N = 64$, $\phi = 5$, $\sigma^2 = 0.01$, $\ell = 0.5$, 1000 unit times $t$). The decreasing slopes of the STR with increasing sample area[12] (**c**), and the SAR with increasing sample duration[12] (**d**) for various communities and in simulations (**g** and **h** $N = 256$, $\phi = 10$, $\sigma^2 = 0.01$, $\ell = 0.5$, spatial window $\Delta A$, temporal windo $\Delta T$). In (**c**) and (**d**) we have rescaled the sample area/duration by the smallest/shortest reported value and coloured by community (see original study for details). In (**g**) and (**h**) we study the STAR in metacommunities of various size $N$, represented by colour. Limited spatio-temporal turnover in the smallest metacommunties (blue colours) greatly reduces the exponents of the STAR relative to large metacommunities (red colours). $A_{ij} = 0.5$ with probability 0.5 in all cases.

*Community structure.* Temporal turnover has been posited to play a stabilising role in the maintenance of community structure[57,58]. In an estuarine fish community[59], for example, species richness (Fig. 6b) and the distribution of abundances were remarkably robust despite changes in population biomasses by multiple orders of magnitude. In model metacommunities with autonomous turnover we found, likewise, that local species richness exhibited only small fluctuations around the steady-state mean (Fig. 6f, three random local communities shown) and that the macroscopic structure of the community was largely time invariant (Supplementary Fig. 12). In the light of our results, we propose the absence of temporal change in community properties such as richness or the abundance distribution despite potentially large fluctuations in population abundances[59] as indicative of autonomous compositional turnover.

*The species-time-area-relation, STAR.* The species-time-relation (STR), typically fit by a power law of the form $S \propto T^w$ [12,60,61], describes how observed species richness increases with observation time $T$. The exponent $w$ of the STR has been found to be consistent across taxonomic groups and ecosystems[12,13,62], indicative of some general population dynamical mechanism. However, the exponent of the STR decreases with increasing sampling area[12], and the exponent of the empirical Species Area Relation (SAR) ($S \propto A^z$) consistently decreases with increasing sampling duration[12] (Fig. 6c, d). We tested for these patterns in a large simulated metacommunity with $N = 256$ patches by computing the species-time-area-relation (STAR) for nested subdomains and variable temporal sampling windows (see "Methods"). We observed exponents of the nested SAR in the range $z = 0.02$–$0.44$ and for the STR a range $w = 0.01$–$0.44$ (Supplementary Fig. 13). We also found a clear decrease in the rate of species accumulation in time as a function of sample area and vice-versa (Fig. 6g, h), consistent with the empirical observations. Meta-analyses of these patterns in nature have reported exponents which are remarkably

consistent, with $z$ typically in the range 0.1–0.3[63], and $w$ typically in the range 0.2–0.4[13], in both cases largely independent of location or taxonomic group[13].

Thus, the distribution of temporal occupancy, the time invariance of key macroecological structures and the STAR in our model metacommunities match observed patterns. This evidence suggests that such autonomous dynamics cannot be ruled out as an important driver of temporal compositional change in natural ecosystems.

*Turnover rate in simulated metacommunities.* How do the turnover rates that we find in our model compare with those observed? Our current analytic understanding of autonomous turnover is insufficient for estimating the rates directly from parameters, but the simulation results provide some indication of the expected order of magnitude, that can be compared with observations. Key for such a comparison is the fact that, because the elements of **R** are 1 on average, the time required for an isolated single population to reach carrying capacity is $\mathcal{O}(1)$ unit times. Supplementary Fig. 12b suggests that transitions between community states occur at the scale of around 10–50 unit times. This gives a holistic, rule-of-thumb estimate for the expected rate of autonomous turnover, depending on the typical reproductive rates of the guild of interest. In the case of macroinvertebrates, for example, the time required for populations to saturate in population biomass could be of the order of a month or less. By our rule of thumb, this would mean that autonomous community turnover would occur on a timescale of years. In contrast, for slow growing species like trees, where monoculture stands can take decades to reach maximum population biomass, the predicted timescale for autonomous turnover would be on the order of centuries or more. Indeed, macroinvertebrate communities have been observed switching between community configurations with a period of a few years[64,65], while the proportional abundance of tree pollen and tree fern spores fluctuates in rain forest bog

deposits with a period of the order of $10^3$ years[66]—suggesting that the predicted autonomous turnover rates are biologically plausible.

**Conclusions.** Current understanding of the mechanisms driving temporal turnover in ecological communities is predominantly built upon phenomenological studies of observed patterns[2,67–69] and is unquestionably incomplete[10,59]. That temporal turnover can be driven by external forces—e.g., seasonal or long term climate change, direct anthropogenic pressures—is indisputable. A vitally important question is, however, how much empirically observed compositional change is actually due to such forcing. Recent landmark analyses of temporal patterns in biodiversity have detected no systematic change in species richness or structure in natural communities, despite rates of compositional turnover greater than predicted by stochastic null models[1,70–72]. Here we have shown that empirically realistic turnover in model metacommunities can occur via precisely the same mechanism as that responsible for regulating species richness at the local scale. While the processes regulating diversity in natural communities remain insufficiently understood, our theoretical work suggests local structural instability may explain these empirical observations in a unified and parsimonious way. Therefore, we advocate for the application of null models of metacommunity dynamics that account for natural turnover in ecological status assessments and predictions based on ancestral baselines. Future work will involve fitting the model described here to observations by estimating abiotic and biotic parameters from empirical datasets. In the Supplementary Discussion we show how different combinations of parameters lead to different quantitative outcomes (Supplementary Fig. 4), likely representing different types of empirical metacommunities. Understanding where in this parameter space natural systems exist may provide the foundation for a quantitative null model, a baseline expectation of turnover against which observations can be compared.

Our simulations revealed a qualitative transition from "small" metacommunities, where autonomous turnover is absent or minimal, to "large" metacommunities with pronounced autonomous turnover (Fig. 2). The precise location of the transition between these cases depends on details such as dispersal traits, the ecological interaction network, and environmental gradients (Supplementary Fig. 4). Taking, for simplicity, regional species richness as a measure of metacommunity size suggests that both 'small' and 'large' communities in this sense are realised in nature. In our simulations, the smallest metacommunities sustain 10s of species, while the largest have a regional diversity of the order $10^3$, which is not large comparable to the number of tree species in just $0.25\,km^2$ of tropical rainforest (1100–1200 in Borneo and Ecuador[73]) or of macroinvertebrates in the UK (>32,000[74]). Within the 'small' category, where autonomous turnover is absent, we would therefore expect to be, e.g., communities of marine mammals or large fish, where just a few species interact over ranges that can extend across entire climatic niches, implying that the effective number of independent "patches" is small and providing few opportunities for colonisation by species from neighbouring communities. Likely to belong to the 'large' category are communities of organisms that occur in high diversity with range sizes that are small compared to climatic niches, such as macroinvertebrates. For these, autonomous turnover of local communities can plausibly be expected based on our findings. Empirically distinguishing between these two cases for different guilds will be an important task for the future.

For metacommunities of intermediate spatial extent, autonomous turnover is characterised by sharp transitions between cohesive states at the local scale. To date, few empirical analyses

have reported such coherence in temporal turnover, perhaps because the taxonomic and temporal resolution required to detect such patterns is not yet widely available. Developments in biomonitoring technologies[75] are likely to reveal a variety of previously undetected ecological dynamics, however and by combining high resolution temporal sampling and metagenetic analysis of community composition, a recent study demonstrated cohesive but short-lived community cohorts in coastal plankton[76]. Such Clementsian temporal turnover may offer a useful signal of autonomous compositional change in real systems.

Thus, overcoming previous computational limits to the study of complex metacommunities[11,77], we have discovered the existence of two distinct phases of metacommunity ecology— one characterised by weak or absent autonomous turnover, the other by continuous compositional change even in the absence of external drivers. By synthesising a wide range of established ecological theory[11,23,24,47–49], we have heuristically explained these phases. Our explanation implies that autonomous turnover requires little more than a diverse neighbourhood of potential invaders, a weak immigration pressure, and a complex network of interactions between co-existing species.

## Methods

**Metacommunity assembly.** The dynamics of local population biomasses $B_{ix}(t)$ were modelled using a spatial extension to the multispecies Lotka-Volterra competition model[23]:

$$\frac{dB_{ix}}{dt} = B_{ix}\left(R_{ix} - \sum_{j=1}^{S} A_{ij}B_{jx}\right) - e\,B_{ix} + \sum_{y\in\mathcal{N}(x)} \frac{e}{k_y}\exp\left(-d_{xy}\ell^{-1}\right)B_{iy}. \quad (2)$$

The competitive coupling coefficients $A_{ij}$ for $i \neq j$ were sampled from discrete distributions. Generally, $A_{ij}$ were set to 0.5 with a probability of 0.5 and to 0 otherwise, however, for the simulation shown in Fig. 5, we relaxed the dynamic coupling and instead set $A_{ij}$ to 0.3 with a probability of 0.3. This delayed the onset of local structural instability during metacommunity assembly, making the coincident emergence of local biodiversity regulation and autonomous compositional turnover visually clearer.

Environmental heterogeneity was modelled implicitly through spatial variation in species' intrinsic growth rates $R_{ix}$. Specifically, the $R_{ix}$ were sampled independently for each species $i$ from a Gaussian random field[78] with mean $\mu = 1.0$ and standard deviation $\sigma$, generated via spectral decomposition[79] of the $N \times N$ landscape covariance matrix with elements $\Sigma_{xy} = \exp[-\phi^{-1}d_{xy}]$, where $d_{xy}$ denotes the Euclidean distances between patches $x$ and $y$, and $\phi$ the autocorrelation length (Supplementary Fig. 3).

The dispersal matrix $\mathbf{D}$ (Eq. (1)) has diagonal elements $D_{xx}$ of $-e$, where $e$, the fraction of biomass leaving patch $x$ per unit time, was kept fixed at 0.01 for all simulations. For pairs of patches connected by an edge in the spatial network, the immigration terms were modelled as negative exponentials $D_{xy} = ek_y^{-1}\exp(-d_{xy}\ell^{-1})$, controlled by a dispersal length parameter $\ell$, thus assuming a propensity for propagules to transition to nearby sites. The normalisation constant $k_y$ divides the biomass departing patches $y$ between all other patches in its local neighbourhood ($\mathcal{N}(y)$), weighted by the ease of reaching each patch i.e., $k_y = \sum_{z\in\mathcal{N}(y)}\exp(-d_{yz}\ell^{-1})$, implying an active dispersal process.

Metacommunities were assembled by iterated regional invasion (Supplementary Fig. 1). In each iteration of the algorithm, $0.05S + 1$ new species were introduced to the metacommunity, with $S$ denoting the current extant species richness. The invaders were tested to ensure positive growth rates at low abundance. This was done by introducing a multiple of $0.05S + 1$ newly generated species into all patches at very low abundance, then simulating for a handful of time steps and testing for increasing biomass trajectories in at least one patch. Of the successful invaders, $0.05S + 1$ were randomly selected and each introduced at $10^{-6}$ biomass units into the patch in which its growth rate was greatest during testing. After invaders were introduced, metacommunity dynamics were simulated using the SUNDIALS[80] numerical ODE solver. The time between regional invasions we kept fixed at 500 unit times, and before each new regional invasion the metacommunity was scanned and species with biomass smaller than $10^{-4}$ biomass units in all patches of the network were considered regionally extinct and removed from the model. The assembly algorithm aims to remove all species whose total biomass declines to zero in the course of the system's complex dynamics. In rare cases autonomous fluctuations may drive one of the remaining species to very low abundance in all patches, however the majority retain local biomass above the detection threshold in at least one patch at all times.

To assemble models of sufficient spatial extent and species richness, we developed a parallel implementation of the assembly model that makes use of the

algorithmic domain decomposition method[81] for the population-dynamical simulations. This involves decomposing the metacommunity into spatial subdomains of equal numbers of patches, each of which is simulated by a unique parallel process (CPU), with boundary states regularly broadcast between processes. The code was run on the Apocrita high-performance cluster at Queen Mary, University of London[82]. This permitted assembly of saturated metacommunities of up to $N = 256$ patches harbouring $S \sim 3000$ species, thus breaking through frequently lamented computational limits[11,77] on the numerical study of metacommunities.

**Quantifying autonomous turnover.** All post-assembly analyses were done using the R statistical software environment[83] (version 4.0.3). For fully assembled metacommunities, we simulated and stored time series of $t_{max} = 10^4$ metacommunity samples $B_{ixt} = B_{ix}(t)$ taken in intervals of one unit time. In these metacommunity timeseries, we measured spatio-temporal turnover based on (i) compositional dissimilarity, (ii) the distribution of temporal occupancy, (iii) the number of compositional clusters detected using hierarchical clustering, and (iv) via species accumulation curves generated using sliding spatial and temporal sampling windows. Metrics were selected in order to answer specific questions, or for comparison to observed patterns. Some analyses require quantifying local species richness. This was done by setting a detection threshold of $10^{-4}$ biomass units, below which populations are considered absent from the community. Local source diversity, which we define in Fig. 5, is a related but different diversity measure that is more adequate for quantifying the component of a local community subject to local ecological limits to biodiversity.

*Compositional dissimilarity.* Spatial/temporal compositional dissimilarity was quantified using the Bray-Curtis[42] index via the function `vegdist` in the R package "vegan"[84].

*Temporal occupancy.* We assessed temporal occupancy by first converting biomass into presence-absence data ($P_{ixt} = 1$ for all $B_{ixt} > 10^{-4}$, and 0 otherwise). Then, for all populations present at least once, we computed the temporal occupancy ($TO_{ix}$) as the proportion of the time interval of length $t_{max}$ during which that population was present:

$$TO_{ix} = \frac{1}{t_{max}} \sum_t P_{ixt} \quad (3)$$

*Hierarchical clustering.* We assessed the degree of temporal clustering in community composition using complete linkage hierarchical clustering[85] of the Bray-Curtis dissimilarity matrix, which gives an approximate measure of the number of unstable equilibria between which the dynamical system fluctuates. We computed the number of clusters using a threshold of 25% dissimilarity, which reflects the structure visible in pairwise dissimilarity matrices (Supplementary Fig. 6a, b).

*Spatio-temporal species accumulation.* We studied the STR and SAR in model metacommunities using a sliding window approach, asking, for given $\Delta A \in \mathbb{N}$ and $\Delta T \in \mathbb{R}^{>0}$, how many species $S^{obs}$ were detected on average in sets $\mathcal{A}$ of $\Delta A = |\mathcal{A}|$ patches during any time interval $\mathcal{T}$ of $\Delta T$ unit times length. Specifically, for a metacommunity of $N = 2^8 = 256$, the spatial windows were $\Delta A \in \{2^0, 2^1, \ldots, 2^8\}$ patches, while the temporal windows were $\Delta T \in \{1, 5, 10, 50, 100, 500, 1000\}$ unit times. For each patch $x \in \{1, \ldots, N\}$ the spatial sub-sample was then defined as the set $\mathcal{A}$ consisting of the focal patch and its $\Delta A - 1$ nearest neighbours. Similarly, for each $t \in \{1, \ldots, t_{max} - \Delta T\}$ the sliding temporal window $\mathcal{T}$ was defined as the $\Delta T$ successive recording time steps in the range $t$ to $t + \Delta T$. The species richness observed in a given spatio-temporal sub-sample was then computed as

$$S^{obs} = \sum_i \left[ \sum_{t \in \mathcal{T}} \sum_{x \in \mathcal{A}} P_{ixt} \geq 1 \right], \quad (4)$$

where the Iverson brackets [.] denote the indicator function ensuring species are counted only once. Finally, the average of $S^{obs}$ for a given spatio-temporal sample size was computed in all combinations of $\Delta A$, $\Delta T$.

In closed systems, species accumulation in both space and time must ultimately saturate, either when the entire metacommunity or entire time series is sampled. Thus we defined the exponents $z$ and $w$ of the STAR as the maximum slopes of the SAR/STR on double logarithmic axes (Supplementary Fig. 13).

**Reporting summary.** Further information on research design is available in the Nature Research Reporting Summary linked to this article.

## Data availability

Simulation data supporting a subset of the results of this study are available at https://doi.org/10.6084/m9.figshare.14139644.v1.

## Code availability

The software used to generate these data is available at https://github.com/jacobosullivan/LVMCM_src[86].

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

## Acknowledgements

We thank Lars Chittka, and Laurent Frantz for their helpful comments on earlier drafts of this paper. This work forms part of the project "Mechanisms and prediction of large-scale ecological responses to environmental change" funded by the Natural Environment Research Council (NE/T003510/1).

## Author contributions

A.G.R. conceived of the study. J.D.O. and A.G.R. designed the model. J.D.O. developed the model, performed simulations, analysed the data and drafted the manuscript. J.D.O., C.D.T. and A.G.R. all interpreted model outputs in comparison with observations and contributed to manuscript writing.

## Competing interests

The authors declare no competing interests.
