## [Peer Review File · Nature Communications]

Reviewer comments, first round –

Reviewer #1 (Remarks to the Author):

The Authors provide an important assessment of how metacommunities dynamics may underpin changes in composition over time in the absence of environmental change or gradients. The study provides a thorough assessment of extensive simulations to detect and quantify turnover due solely to ecological dynamics, and shows how such autonomous turnover can emerge as the spatial extent of metacommunities increases. The findings illustrate that a combination of high richness of potential invaders with a constant weak flow of immigration, upon a network of connected patches with competing species can lead to such autonomous turnover. The Authors further show that their models are able to replicate several crucial empirical patterns, including macroecological relationships. The study represents a very timely and relevant contribution to a particularly important area of current research – particularly, given the ongoing debate about how biodiversity is changing in the current situation of increased anthropogenic pressures, alongside recent analyses showing regulation of some community-level metrics (such as richness and abundance). Temporal turnover has been shown to be the major signature of biodiversity change over recent decades, but what drives such changes in composition remains yet to be specified. This study illustrates how intuitive concepts from population and metacommunities dynamics can lead to compositional change in the absence of drivers of change, and further highlights differences between local and regional scales – another relevant aspect of the ongoing discussion related to how biodiversity is being lost or affected across scales. Consequently, the study can have important implications for biodiversity research, management and conservation. I believe the study is potentially suitable for Nature Communications, and overall, I found the paper to be very well written. Still, I feel that the overall logic of the paper was heavily dominated by the methodological implementation, and “hiding” the important findings presented, in detriment of a quite relevant contribution, in my opinion.

I provide some comments that I hope the Authors will find useful, and which I hope can improve the manuscript's clarity and impact.

1) Overall, I think the results and their implications are quite important and of likely general interest to many ecologists, but they are somewhat obscured by technical language. While the manuscript is clear and well articulated, there were many instances where the jargon and technical language is too complex and hard to follow, especially if readers are not super statistical-oriented. The methods/models are not easy to grasp unless you are already familiar with the Authors previous work (I guess familiarity with metacommunity concepts is less of an issue). I note that theoretical ecology is not exactly my direct area of study, so I made several comments from the perspective of someone who would be very much interested in understanding the underlying mechanisms of temporal turnover and community patterns.

2) I was left wondering a bit how this contribution differs from the recent paper by the same authors in Ecology Letters (O'Sullivan et al. 2019, Ecol Lett, 22: 1428-1438); some findings are quite similar between the two papers, so I urge the Authors to more explicitly state/show the added relevance of their findings in the current ms.

3) Another general comment is that I think it would be important to precisely define “community turnover” (here it is used in the broadest sense and including changes in both composition and abundance), given the sometimes confusing nomenclature associated with such metrics, and also given the Authors also explore two different components in their supplementary analysis (one of which is also called turnover). Related, I think BC similarity is never defined in the main text, and it seems it is dissimilarity that is being calculated. I also ask the Authors to revise their statements (and analyses if necessary!) regarding the multiple-site metrics in Baselga's partition framework in their Supp. Information (detailed comment below).

4) Related to the above, in several instances the Authors use terms without precisely defining them, e.g. regional diversity, regional scale, steady-state dynamics, ecological structural instability.

5) I understand the constraints of implementing the extensive simulations and analyses, as well as

the need to keep the results relatively simple, but I would suggest discussing potentially relevant implications of using e.g. 3000 species in some simulations, as well as how the time scale and resolution of the simulations relate to empirical data. I make some more detailed comments below regarding this as well.

6) A minor comment is that sometimes important references were missing from the manuscript or perhaps the ones used were not the most adequate; I provide some examples and suggestions below.

Specific comments

Introduction

1. As per my general comment, the Introduction is written rather technically, which is fine for people who work with theoretical ecological models, but perhaps not ideal for a broader readership, including many ecologists. Therefore, I think a very useful purpose this manuscript could serve would be to provide quantitative sound expectations of such models to help ecologists and global change researchers to take the best advantage of them, and to be able to evaluate them in line with empirical analyses.

2. I think the above point is also evident from the quite complex figures presented (very nice figures!, but still complex) – particularly Figs. 1 and 2 might already leave the readers wondering what is being shown. Perhaps a very simple conceptual diagram illustrating the model settings/structure implemented could be added to the ms to set the scene before showing the first results.

3. Lines 18-19: I do not think Vellend et al. is a suitable reference for this statement.

4. Line 28-32: Consider adding reference to the recent study by Blowes et al. 2019, showing strong support for a global signature of turnover across thousands of communities (i.e. using more extensive time-series data); and also adding the recent IPBES report e.g.?

Blowes, S.A. et al. The geography of biodiversity change in marine and terrestrial assemblages. *Science* 366, 339–345 (2019).

5. Lines 33-38: Missing references?

6. Line 39: "So, can communities of many interacting species turn over autonomously?" I would argue that we already know the answer is yes; the issue is how much of turnover can be attributed to "normal underlying processes" versus turnover due to (accelerated) anthropogenic drivers; indeed this is what the Authors mention in the Discussion. I guess it is perfectly fine that the current framework and manuscript are not precisely trying to quantify this, but I think this could be stated more clearly.

7. Lines 51-52: What are "numerical ecological traits"? This sounds vague and not detailed enough at the same time. And where were these traits sampled from?

8. Lines 49-55: I think important details about the simulated metacommunities are missing from the main text; e.g. How many metacommunities were simulated? How is "regional diversity" defined? and does this vary? Does $N=256$ represent this regional scale? How heterogeneous was the spatial abiotic environment implemented?

9. Lines 64-65: Missing references?

10. Line 71: Worth mentioning what "Gabriel graph" is or what purpose it serves?

11. Line 78: "...populations attain equilibria, implying the absence of autonomous turnover..." How is the equilibrium defined? It was mentioned before that the metacommunities were assembled until regional diversity reaches an asymptote, which I assume refers to species richness, but this mentions populations. I think there is a lot of information that needs to be unpacked here (i.e. there are too many steps encapsulated here).

12. Line 83: What are these "compositional states"? This is never really defined or explained in the main text; perhaps add a bit more detail here.

13. Line 99: Consider saying "colonisations" instead of "invasion" to avoid confusion with regional invasion mentioned before?

14. Lines 128-131: I found this sentence hard to follow, too complex and technical; there are so many complex concepts being included here, and at the same time, it seems to be tangential to the results being presented. Could the Authors clarify and/or better integrate these concepts with the overall findings?

15. Lines 132-133: Please specify if this refers to mass effects.

16. Line 150: What drives the variation in species richness among these different patches? Do the patches have different sizes, different environmental heterogeneity, as a function of

dispersal/distance between them? Or are you here referring to higher neighbourhood richness in communities with increasing N? Sorry if I am missing something, but this was not clear to me, i.e. how different patches "reach" different species richness when describing this pattern.

17. Line 164: What is "ecological structural instability", how is this defined?
18. Lines 165-167: Another sentence hard to follow, too complex and technical.
19. Line 171: Also need to specify what is "local structural instability".
20. Lines 173-177: Missing references?
21. Line 178: Misspelled "represented".
22. Line 183: Misspelled "external".
23. Lines 186-198: Could the Authors mention/discuss a bit about the differences in diversity between the simulated and empirical communities shown as examples in Fig.5? How do e.g. species richness or the time-scales compare – would each somehow match the yearly resolution in empirical datasets?
24. Lines 196-198: Perhaps this statement is too strong/broad/general? And could the Authors expand on what they mean by "autonomous ecological dynamics"? Does this refer to species being replaced and/or too changes in the rank abundances?
25. Lines 199-201: Missing references?
26. Line 208: The limits reported for the exponents don't seem to match what is shown in Figs. 5 or S11?
27. Lines 216-218: Perhaps it would make sense to add references to recent analyses quantifying rates of compositional change (here or elsewhere in your discussion), e.g. Blowes et al. 2019 (as before), Pilotto et al. 2020, Burrows et al. 2019.

Pilotto, F. et al. Meta-analysis of multidecadal biodiversity trends in Europe. *Nat. Commun.* 11, 3486 (2020).

Burrows, M.T. et al. Ocean community warming responses explained by thermal affinities and temperature gradients. *Nat. Clim. Chang.* 9, 959–963 (2019).

28. Line 223: Again, I think the reference to Vellend et al. 2013 here might be misused, as they did not evaluate compositional change, correct? And again I would suggest adding Blowes et al. 2019 here, as it reinforces and expands on the Dornelas et al. 2014 study.
29. Line 233: What is a "macroclimatic niche"?
30. Lines 232-237: Would these patterns also be influenced by dispersal ability additionally to species having wide niche space/large ranges?
31. Lines 239-241: Could the Authors mention something about how temporal resolution in their simulated communities compare to that of empirical data? Would it be relevant for seasonal, yearly, decadal analyses? For instance, the data included in the BioTIME database is at a yearly resolution, and I would argue there is a lot of data in this source and elsewhere too (not trying to make a "nasty comment", just wondering how the simulations time "range" and "resolution" would match to empirical data 9)
32. Lines 251-254: Really nice final sentence.

Figures

33. Figure 1: Does the figure correspond to "large model metacommunities" (there is also small N)? Doesn't the figure describe the "Development of autonomous turnover as metacommunities size increase"? Does the figure show BC similarity or dissimilarity? Please indicate here and/or in the text what the range 0-1 means for the similarity (or dissimilarity?) metrics.
34. Figure 2: I think this Figure could be in the Supp. information; it was not super clear to me the difference between the blue and red trajectories in the actual figure without reading the legend.
35. Figure 4: What is " α_{src} "? Also, perhaps need to define/explain spatial turnover here.

Material and Methods

36. Lines 384-385: Suggest adding this detail to main text.
37. Lines 390-391: Does this mean the patches are constantly losing biomass? Why is this implemented/necessary, and how does this relate to populations/communities reaching the equilibrium states?
38. Line 410: "domain decomposition of the spatial network" – what does this mean?
39. Line 441: Please specify what ΔA is, and consider if this can be confused with the competition coefficients notation?

Supplementary text

40. Line 454: So "variability of environment" did not vary?

41. Lines 463-468: This is obviously ideal for implementing the simulations with the purpose of detecting autonomous turnover, but then begs the question of how "specific" or "general" are your results right? And related, is there any sense of how this parameter space is likely to match up/represent empirical communities?

42. Lines 471-474: The multiple-site dissimilarity metrics developed by Baselga are affected by the number of sites involved in the calculation, and this should be controlled for, as the effects on dissimilarity would be driven by the number of sites included (see Baselga 2010). Can the Authors please clarify and check if necessary when using the multi-site metrics (assuming that comparing extents with different number of sites would be similar to comparing neighbourhoods with different number of nodes).

43. Line 586: Missing references?

44. Figure S8: Specify if Bray-Curtis is similarity or dissimilarity; missing figure legend for heat-maps.

45. Figure S9: Specify if Bray-Curtis is similarity or dissimilarity.

Reviewer #2 (Remarks to the Author):

The manuscript by O'Sullivan and colleagues seeks to test whether autonomous, or competition-driven, turnover of species through time can reproduce observed patterns of temporal change in species diversity. The study finds that simple Lotka-Volterra competition models embedded in a metacommunity framework can indeed produce substantial autonomous turnover of species through time once they are made sufficiently large (in their number of local communities) to overcome species' extinctions and community collapse. Rates of temporal turnover of species well match those in empirical studies and spatial scaling of species richness across area and through time are also can match well those of empirical studies.

Overall, I do not find the results too surprising. We know that models with pure stochastic drift and neutral community dynamics (e.g. Hubbell's neutral model) that is more rapid than natural rates of speciation for communities under most circumstances (e.g. except when dispersal is very limited). It is also not surprising to me that a metacommunity model can be tuned to match empirical patterns. In part because the empirical patterns do not contain a large amount of information and could be produced by many possible mechanisms, so a match of patterns at the level conducted seems like weak evidence for the importance of autonomous turnover. Studies such as reference 1 suggest that richness is anchored by something, and the typical explanation is niches. Are the results of the present manuscript not indicating that if we tune a manuscript with appropriate R values, competition coefficients and dispersal rates (and the consequent competition-colonization tradeoffs that are set or emerge in surviving species) we can achieve something like this? I think we have known this for quite some time. For real communities our difficulty is to distinguish environmental covariation of species from actual interactions: negative interspecific covariation in responses to environmental conditions produces patterns that look like interspecific competition. It is also hard to quantify competition for more than a dozen species and even that number creates a large number of combinations of species that could interact.

The arguments in the two paragraphs of lines 26-38 that we can show that autonomous species interactions could be responsible for temporal change in communities seem overly simplistic. We fully expect that there are environmental drivers of change in communities: a large literature on disturbance supports this. Also we expect that species interactions can drive dynamics, a large literature on species interactions and ecological succession support this. It is not clear to me how the knowledge gained from the current work would help an ecologist determine that environmental drivers are unimportant or that species interactions are.

The metacommunity model is described to have local variation in R, so that spatial variation in the environment is included but the local values of R are presumably fixed through time. Species dispersing across space creates a move from $R_{i,x1}$ to $R_{i,x2}$ so the species change their R-values

through dispersal. Consequently, the amount of variation in R-values seems important to the idea that species turnover is purely autonomous. This at least needs a clearer explanation if not more work to determine its importance.

Reviewer #3 (Remarks to the Author):

This is an intriguing work that tries to show us only species interaction and dispersal can exhibit turnover pattern observed in nature, providing us an alternative explanation other than environmental change and stochasticity. This work presented interesting results and mechanistic explanations. The results of the meta-communities and isolated LV systems were well matched. However, there are still several aspects I feel the authors might need to work on. 1) The introduction can be re-edited to make it more smooth and trackable, mainly introducing the patterns the authors want to test in the results. 2) The methods need more details about the simulation. 3) some results need more clarification. My specific comments are listed below.

A major concern is the writing style of Introduction. I found it difficult to follow the flow of MS. Especially confused by what it is expected to see in the results. Therefore, in the introduction, maybe introduce what the readers expect to see about the turnover patterns observed in real systems and also list those they want to compare with their simulation.

Fig. 1B, it is the pattern in a local community. Do all local communities have a similar pattern? Is it a typical one? Or they are showing a specific case? My concern is what the authors wanted to show for reader, the temporal pattern in single local communities or meta-communities? Is the transition of turnover pattern prevalent in empirical meta-community? I am asking this because the authors declare the simulated patterns are observed in natural systems. What does the color gradient mean in Fig. 1C? Which part of text corresponds to Fig. 1C?

L186: Temporal occupancy is TO_{ix} . What is the data for the density distribution in Fig. 5A,E. Is it TO_i averaged over all sites or TO_x averaged over local species or TO_{ix} ?

L 214: Fig. 5. All of a sudden, the readers see the comparison between bird and simulated community, then fish data were presented. Why different dataset? First, it is better to introduce them in introduction. Second, one can often find a specific dataset with similar pattern to a model; my question is how a single example can prove that the pattern observed in model community is commonly observed in natural communities? or the authors just want to show a similar empirical pattern as in theoretical model?

L214: Fig. 5CD, are the data the same with ref.39? maybe mention that it is redrawn from ref.39.

L228: The theoretical model shows empirical turnover pattern might emerge from just species interaction and dispersal. But I find it uneasy to apply it as null model because the parameters of the theoretical model like competition coefficient, dispersal rates are not easy to get. Otherwise, one has to randomly draw parameters, but this adds stochasticity to the model.

Technique points:

The methods section needs more details.

L63: "spatially correlated normal distribution". Could they explain more about ϕ and σ^2 , give the formulation? Or give a reference or package for this.

L377: pls give the integrator for the equation S1 somewhere in the methods

L398-408: The assembly of meta-communities needs more clarification and details. I am not sure if I get it right. In each simulation, did they invade the meta-community with $0.05S + 1$ first, then numerically solve the equation S1 from t to $t+1$, then invade again, until $t=1e+4$. Then use the meta-community data from $t=0$ to $t=1e+4$ to analyze temporal beta diversity? Or species invade, then solve S1 from $t=1$ to $t=1e+4$, then invasion, solve S1 from $t=1$ to $t=1e+4$, repeat it until intrinsic limit on the number of species regionally. Or apply the assembly process first, then numerically solve the equation S1 from $t=1$ to $t=1e+4$? Please explain the simulation in detail, because this is the core method. A workflow plot or pseudo codes will be more clear. Since the

codes are not provided, it is crucial for readers to know exactly how the authors simulated the meta-community dynamics.

L400: "The invaders were tested to ensure positive growth"? How do they test it? The invading species was introduced to all local sites, or a random site?

L427: I find the terminology is sometimes confusing. In methods, it is termed spatio-temporal compositional similarity. It seems to me a novel measure, but just indicates spatial/temporal similarity.

L402: "The metacommunity was periodically scanned", what is the period/time interval.

L469, and Figure S9: Pls explain this "the time averaged spatial community dissimilarity at the local neighbourhood scale" in detail. How can a community dissimilarity index represent the three environmental parameters? What is the link between "the time averaged spatial community dissimilarity at the local neighbourhood scale" and the three parameters? What does each data point in Fig. S9 represent?

Response to reviewer comments

We are very much in debt to all three reviewers for their critical reading of our manuscript and the supportive, detailed and stimulating feedback they provided.

Below please find *the original reviewer comments in black italic* and our responses to those comments in blue. In the revised manuscript, all new or modified text is marked in blue. Line numbers in our response to the reviewers below refer to the revised manuscript.

Reviewers' comments:

Reviewer #1 (Remarks to the Author):

The Authors provide an important assessment of how metacommunities dynamics may underpin changes in composition over time in the absence of environmental change or gradients. The study provides a thorough assessment of extensive simulations to detect and quantify turnover due solely to ecological dynamics, and shows how such autonomous turnover can emerge as the spatial extent of metacommunities increases. The findings illustrate that a combination of high richness of potential invaders with a constant weak flow of immigration, upon a network of connected patches with competing species can lead to such autonomous turnover. The Authors further show that their models are able to replicate several crucial empirical patterns, including macroecological relationships. The study represents a very timely and relevant contribution to a particularly important area of current research – particularly, given the ongoing debate about how biodiversity is changing in the current situation of increased anthropogenic pressures, alongside recent analyses showing regulation of some community-level metrics (such as richness and abundance). Temporal turnover has been shown to be the major signature of biodiversity change over recent decades, but what drives such changes in composition remains yet to be specified. This study illustrates how intuitive concepts from population and metacommunities dynamics can lead to compositional change in the absence of drivers of change, and further highlights differences between local and regional scales – another relevant aspect of the ongoing discussion related to how biodiversity is being lost or affected across scales. Consequently, the study can have important implications for biodiversity research, management and conservation. I believe the study is potentially suitable for Nature Communications, and overall, I found the paper to be very well written. Still, I feel that the overall logic of the paper was heavily dominated by the methodological implementation, and “hiding” the important findings presented, in detriment of a quite relevant contribution, in my opinion.

I provide some comments that I hope the Authors will find useful, and which I hope can improve the manuscript's clarity and impact.

1) Overall, I think the results and their implications are quite important and of likely general interest to many ecologist, but they are somewhat obscured by technical language. While the manuscript is clear and well articulated, there were many instances where the jargon and technical language is too complex and hard to follow, especially if readers are not super statistical-oriented. The methods/models are not easy to grasp unless you are already familiar with the Authors previous work (I guess familiarity with metacommunity concepts is less of an issue). I note that theoretical ecology is not exactly my direct area of study, so I made several comments from the perspective of someone who would be very

much interested in understanding the underlying mechanisms of temporal turnover and community patterns.

In response to the reviewer's comments we have completely re-written the introduction. While the updated version retains many of the key ideas of the previous version, we have aimed to make it more accessible and emphasise points of broader interest. We have also restructured the manuscript such that a thorough (but largely non-technical) description of the model is now part of the results section, rather than the introduction. We believe this is justified in the sense that the development of a highly customizable metacommunity model which draws heavily on novel computational developments is a result in and of itself, and also find that with this structure the introduction becomes much more focused on the ecological questions our model addresses.

2) I was left wondering a bit how this contribution differs from the recent paper by the same authors in Ecology Letters (O'Sullivan et al. 2019, Ecol Lett, 22: 1428-1438); some findings are quite similar between the two papers, so I urge the Authors to more explicitly state/show the added relevance of their findings in the current ms.

To clarify, novel computational developments permit the assembly of much larger (in terms of number of patches) and more speciose metacommunities. This allowed, for the first time, a detailed study of the parameter space in which autonomous turnover occurs. We have added the following sentence to line 53-55 to clarify and emphasise the key developments this study introduces:

"Here we build upon this work by exploring the spatio-temporal patterns that emerge in metacommunity models. As shown below, these arise when expanding the spatial and taxonomic scale of simulations beyond those studied previously."

3) Another general comment is that I think it would be important to precisely define "community turnover" (here it is used in the broadest sense and including changes in both composition and abundance), given the sometimes confusing nomenclature associated with such metrics, and also given the Authors also explore two different components in their supplementary analysis (one of which is also called turnover). Related, I think BC similarity is never defined in the main text, and it seems it is dissimilarity that is being calculated. I also ask the Authors to revise their statements (and analyses if necessary!) regarding the multiple-site metrics in Baselga's partition framework in their Supp. Information (detailed comment below).

In our updated introduction we have added a paragraph which we hope now clearly defines autonomous turnover as we use the term (lines 20-28). After discussing the emergence of autonomous population dynamics we write:

"We refer to as 'autonomous turnover' local compositional changes involving significant restructuring of relative abundances or colonisation-extinction processes, driven by autonomous population dynamics" (lines 26-28).

4) Related to the above, in several instances the Authors use terms without precisely defining them, e.g. regional diversity, regional scale, steady-state dynamics, ecological structural instability.

Local/regional diversity:

We have clarified the use of these terms as follows:

“... *local diversity*, the number of species coexisting in a given patch, and the *regional diversity*, the total number of species in the metacommunity ...” (line 76-77).

We have also given more detail on the determinants of local species richness:

“local community dynamics and therefore local limits on species richness depend on a combination of abiotic and biotic *filtering* (non-uniform responses of species to local environmental conditions) (Tilman 1982, Leibold et al. 1982, Chase and Leibold 2003) and immigration from adjacent patches, generating so called *mass effects* in the local community (Holt 1993, Mouquet and Loreau 2002, 2003)” (line 96-99).

Steady state dynamics:

In order to avoid confusion we have removed references to ‘steady state dynamics’ and replaced it with ‘persistent dynamics’ which we define simply by contrast to fixed points:

“Previous theoretical and experimental studies have shown how specific motifs in competitive ecological networks can lead to population abundances which do not arrive at fixed points. Instead, such systems can manifest persistent dynamics which we refer to here as ‘autonomous’ since they do not depend on variation in the external environment or other extrinsic drivers” (line 20-23).

Ecological structural instability:

In the new subsection in “Results” we have included a paragraph formally defining ecological structural stability, and the role it plays in regulating species richness at community and metacommunity scales (line 83-95).

5) I understand the constraints of implementing the extensive simulations and analyses, as well as the need to keep the results relatively simple, but I would suggest discussing potentially relevant implications of using e.g. 3000 species in some simulations, as well as how the time scale and resolution of the simulations relate to empirical data. I make some more detailed comments below regarding this as well.

We have added the following to the discussion in order to give more empirical context to the number of species included in the largest simulations studied:

“Taking, for simplicity, regional species richness as a measure of metacommunity size suggests that both ‘small’ and ‘large’ communities in this sense are realised in nature. In our simulations, the smallest metacommunities sustain 10s of species, while the largest have a regional diversity of the order 10^3 , less than, for example, the number of tree species in just 0.25 km² of tropical rainforest (1,100-1,200 in Borneo and Ecuador, Wright 2002) or of macroinvertebrates in the UK (>32,000, Curson et al. 2019)” (line 320-325)

We discuss the question of model time scales below.

6) A minor comment is that sometimes important references were missing from the manuscript or perhaps the ones used were not the most adequate; I provide some examples and suggestions below.

*Specific comments**Introduction*

1. *As per my general comment, the Introduction is written rather technically, which is fine for people who work with theoretical ecological models, but perhaps not ideal for a broader readership, including many ecologists. Therefore, I think a very useful purpose this manuscript could serve would be to provide quantitative sound expectations of such models to help ecologists and global change researchers to take the best advantage of them, and to be able to evaluate them in line with empirical analyses.*

While quantitative estimates of the rate of autonomous turnover would require both further analytic developments and a model fitting procedure, we can make holistic arguments about the rate of turnover we expect for a particular empirical system. These are based on the observation that populations in isolation (i.e. in the absence of competitive interactions or immigration) reach their carrying capacity over times of order $O(1)$ unit times (t). So. for macroinverts 1 unit time could correspond to a month or less, while for trees it would be of the order of decades. Our results suggest that autonomously fluctuating communities transition between community states in <50 unit times (Fig S11B) giving a rough estimate of the empirical expectation for a given guild. While this is far from a rigorous prediction, we find that the resulting estimates of turnover time are biologically plausible. We have added a paragraph to the discussion that summarises these arguments with real world examples (line 298-315).

2. *I think the above point is also evident from the quite complex figures presented (very nice figures!, but still complex) – particularly Figs. 1 and 2 might already leave the readers wondering what is being shown. Perhaps a very simple conceptual diagram illustrating the model settings/structure implemented could be added to the ms to set the scene before showing the first results.*

We have added two new figures to the manuscript which we hope will clarify the modelled approach implemented. The new Fig. 1 summarises the three key elements of the model - the spatial network, the abiotic gradients, and the local community dynamics which depend on growth rates, interaction coefficients and dispersal. In this figure we also show the time series of five species in three nodes, which we hope will help clarify further the relationship between autonomous population dynamics in heteroclinic networks and community scale compositional turnover demonstrated in Fig 2.

The second is a cartoon diagram added to the Supplementary Material - Fig S1 - which we hope will address any potential confusion regarding the process of assembling a model metacommunity. Since we cannot presuppose familiarity with community or indeed metacommunity assembly modelling, this diagram highlights the iterative, bottom-up nature of the assembly processes and the resulting emergence of multi-scale intrinsic diversity regulation.

We have also added the following to lines 81-82:

“In these metacommunities we ... studied the phenomenology of autonomous community turnover in the absence of regional invasions or abiotic change.”

The fact that the composition of the regional community and the abiotic environment are not changing is a crucial aspect of our results. We hope this note will further clarify the context in which autonomous compositional turnover was studied.

3. Lines 18-19: *I do not think Vellend et al. is a suitable reference for this statement.*

Agreed, this citation was relevant to an earlier version of this sentence. We have removed it.

4. Line 28-32: *Consider adding reference to the recent study by Blowes et al. 2019, showing strong support for a global signature of turnover across thousands of communities (i.e. using more extensive time-series data); and also adding the recent IPBES report e.g.? Blowes, S.A. et al. The geography of biodiversity change in marine and terrestrial assemblages. Science 366, 339–345 (2019).*

We have now cited Blowes (2019) in the updated introduction (line 17).

5. Lines 33-38: *Missing references?*

Text removed.

6. Line 39: *“So, can communities of many interacting species turn over autonomously?” I would argue that we already know the answer is yes; the issue is how much of turnover can be attributed to “normal underlying processes” versus turnover due to (accelerated) anthropogenic drivers; indeed this is what the Authors mention in the Discussion. I guess it is perfectly fine that the current framework and manuscript are not precisely trying to quantify this, but I think this could be stated more clearly.*

This sentence has now been removed from the introduction. In its place we pose the question “can community dynamics enabled by spatial structure account for the observed macroecological patterns in population turnover?” (line 48-49). We believe this improves the framing of study and its novelty.

The question then links to our order-of-magnitude quantification of predicted autonomous turnover rates, discussed under point 1 above.

7. Lines 51-52: *What are “numerical ecological traits”? This sounds vague and not detailed enough at the same time. And where were these traits sampled from?*

Agreed, this language was unclear and has been removed. The new section “Metacommunity model and asymptotic community assembly” should now give a clearer idea of the methodology (lines 57-135).

8. Lines 49-55: *I think important details about the simulated metacommunities are missing from the main text; e.g. How many metacommunities were simulated? How is “regional diversity” defined? and does this vary? Does N=256 represent this regional scale? How heterogeneous was the spatial abiotic environment implemented?*

Again see lines 57-135 and the new Figs. 1 and S1.

9. Lines 64-65: *Missing references?*

We have added a citation to the seminal study on the emergence of oscillatory population dynamics in experimental predator-prey systems (Huffaker 1958) (line 104).

10. Line 71: Worth mentioning what “Gabriel graph” is or what purpose it serves?

For clarity we have added the following here: “... a Gabriel graph, a planar graph generated by an algorithm that, on average, links each local community to four close neighbours. Avoidance of direct long-distance dispersal and the sparsity of the resulting dispersal matrix permit use of efficient numerical methods” (line 115-118).

11. Line 78: “...populations attain equilibria, implying the absence of autonomous turnover...” How is the equilibrium defined? It was mentioned before that the metacommunities were assembled until regional diversity reaches an asymptote, which I assume refers to species richness, but this mentions populations. I think there is a lot of information that needs to be unpacked here (i.e. there are too many steps encapsulated here).

Again, we hope that the additional figure S1 will help clarify what is referred to by *diversity equilibria*, or *limits*, that are a basic assumption of our modelling approach. We hope the new paragraph on ecological structural instability (see above) will also help in this regard.

With regards to population level equilibria, we have added the clarification “... attain equilibria, i.e. converge to fixed points...” (line 139).

It is critical that the distinction between diversity equilibria which occur at both local and regional scales, and biomass equilibria in the sense of dynamical fixed points, is sufficiently clear. We hope that this is now that case but will make any further requested edits/clarifications to ensure this is fully understood by the reader.

12. Line 83: What are these “compositional states”? This is never really defined or explained in the main text; perhaps add a bit more detail here.

We have now removed the phrase ‘compositional states’ and replaced it with ‘compositional clusters’ or in places ‘community states’. In practice what we are referring to with this terminology is clusters in community state space detected using hierarchical clustering.

13. Line 99: Consider saying “colonisations” instead of “invasion” to avoid confusion with regional invasion mentioned before?

Following the reviewer’s recommendation we have changed ‘invasion’ to ‘colonisation’ in all cases where this refers to local events, retained the term ‘invasion’ when the scale is explicitly regional, and replaced the phrase ‘invasion pressure’ which occurs at the local, or local neighbourhood scale, with ‘propagule pressure’.

14. Lines 128-131: I found this sentence hard to follow, too complex and technical; there are so many complex concepts being included here, and at the same time, it seems to be tangential to the results being presented. Could the Authors clarify and/or better integrate these concepts with the overall findings?

This distinction between the concepts of ‘multiple attractors’ and ‘multiple equilibria linked through a heteroclinic network’ is formally very important as the phenomenon we observed, and the mechanism we find to best explains it, are mathematically distinct from

those characterized by Bunin (2017) as a 'Multiple attractor'. However, we agree this level of detail interrupts the flow of the text. We have therefore moved this sentence into a footnote (line 467-469).

15. Lines 132-133: Please specify if this refers to mass effects.

According to the definition of mass effects as “mechanisms for spatial dynamics in which there is net flow of individuals created by differences in population size (or density) in different patches (Shmida & Wilson 1985, Leibold et al. 2004)” we consider the propagule pressure to be formally equivalent to mass effects with the caveat that the source populations are not explicitly modelled, only the rate of arrival in biomass units comes into the mathematics. We clarified this in line 194-196:

“If one now adds to such isolated LV models terms representing weak propagule pressure for all S species (Eq. S5), dynamically equivalent to mass effects occurring in the full metacommunity model (Eq. 1), then none of the S species can entirely go extinct”.

16. Line 150: What drives the variation in species richness among these different patches? Do the patches have different sizes, different environmental heterogeneity, as a function of dispersal/distance between them? Or are you here referring to higher neighbourhood richness in communities with increasing N? Sorry if I am missing something, but this was not clear to me, i.e. how different patches “reach” different species richness when describing this pattern.

No single patch in a metacommunity model is of better or worse underlying quality (including area). The combined randomness of the interaction matrix and growth rate vectors is likely to contribute to the emergent differences in local species richness within a given metacommunity model. We note that our understanding of the determinants of variation in local species richness remains incomplete. However, an important contributing factor is variation in patch connectivity. For Gabriel graphs the average node degree in the limit of infinite N is 4, however the variance in degree is non-zero with nodes at the edges particularly isolated in this sense. We have added the following to make this clearer:

“Thus dynamics within a patch are controlled not by N directly but rather by neighbourhood species richness **which, due to spatial inhomogeneities, varies from patch to patch for metacommunities of a given size N**” (line 212-214).

17. Line 164: What is “ecological structural instability”, how is this defined?

See above.

18. Lines 165-167: Another sentence hard to follow, too complex and technical.

The analytic results summarised in Eqs. S6-12 prove that the threshold between UFP and MA phases discovered by Bunin (2017) is mathematically identical to the structurally unstable limit to species richness discovered by Rossberg (2013). In this sentence we make this point but refer to the supplementary material for the proof. In the main text, we report only the numerical results which confirm the relationship. These are represented in Fig 5.

In order to try to make this point, which is a key result of the present study, more accessible to the reader, it has been re-worded as follows (line 226-232):

“Close inspection of this threshold reveals an important and hitherto unreported relationship between the transition into the MA phase and local ecological limits set by the onset of ecological structural instability, which is known to regulate species richness in LV systems subject to external invasion pressure (Rossberg 2013, O’Sullivan 2019): **in the Supplementary Material we show how the boundary between the UFP and MA phases (Bunin 2017) coincides precisely with the onset of structural instability (Rossberg 2013) (Eqs. S6-S12).** For LVMCM metacommunities, **the relationship revealed analytically in the Supplementary Material** is numerically confirmed in Fig. 5.”

19. Line 171: Also need to specify what is “local structural instability”.

Here we have clarified what is meant by the distinction between local and regional structural instability:

“During assembly, local species richness increases until it reaches the limit imposed by local structural instability. Further assembly occurs via the ‘regionalisation’ of the biota (Ricklefs, 2004) - a collapse in average range sizes (O’Sullivan et al. 2019) and associated increase in spatial beta diversity - until regional diversity limits are reached. The emergence of autonomous turnover coincides with the onset of species saturation *at the local scale*. Autonomous turnover can therefore serve as an indirect indication of intrinsic biodiversity regulation via local structural instability in complex communities” (line 232-238).

20. Lines 173-177: Missing references?

Much of this passage summarised results from the present study. However we have now added citations to Tilman (1982), Leibold (1998) and Chase and Leibold (2003), considered to be seminal discussions of the ‘Species sorting’ paradigm. These help to give context to the claim “In order to invade, however, species need to be capable of passing through biotic and abiotic filters” (line 240-241).

21. Line 178: Misspelled “represented”.

22. Line 183: Misspelled “external”.

Updated.

23. Lines 186-198: Could the Authors mention/discuss a bit about the differences in diversity between the simulated and empirical communities shown as examples in Fig.5? How do e.g. species richness or the time-scales compare – would each *t* somehow match the yearly resolution in empirical datasets?

As discussed above, there is indeed a rule-of-thumb approach to calibrating the unit times employed in numerical simulations with empirical time scales. However, this rule-of-thumb depends strongly on the life history traits of the focal taxa. While it would be possible from these arguments to roughly assess the quantitative agreement of model output and empirical observation, this would require detailed knowledge of the natural systems, and would not be consistent between or indeed within studies in the case of meta-analyses. Local species richness in simulated communities depends strongly on the distribution of

interspecific interaction coefficients, for which we made an arbitrary choice. It is not meant to be comparable with observations. Work to fit a metacommunity model to data by extracting biotic and abiotic parameters from observations is on-going, however at present we employ a pattern orientated validation approach in which we are chiefly concerned with a qualitative comparison of model outcomes to empirical results, and semi-quantitative comparison of dimensionless characteristics, such as scaling exponents.

24. Lines 196-198: Perhaps this statement is too strong/broad/general? And could the Authors expand on what they mean by “autonomous ecological dynamics”? Does this refer to species being replaced and/or too changes in the rank abundances?

Our expectation on the basis of our understanding of autonomous compositional change, is that where communities turnover due to external stressors such as long term environmental change, the macroscopic community structure will also be impacted. While more work is needed to fully explore the qualitative differences in autonomous and externally driven turnover in models, we would generally expect changes in both species richness and evenness as a result of change in environmental conditions, particularly if that change occurs faster than the average regional invasion rate. In the case of the purely autonomous turnover observed in metacommunity models, diversity metrics such as local species richness and rank abundance distributions are largely time invariant. We would propose, therefore, that temporal robustness of community-level properties despite large population-level fluctuations is a signal of autonomous turnover.

In response to the question about “autonomous ecological dynamics” we have changed this to “autonomous compositional turnover”. We have also added in the introduction clarifications of terminology (line 265).

25. Lines 199-201: Missing references?

Preston (1960) first proposed the conceptual equivalence of the Species Area Relation and the Species Time Relation. Important follow-up work was led by Adler (Adler and Lauenroth 2003, Adler et al. 2005). We have added citations to this sentence (line 267).

26. Line 208: The limits reported for the exponents don't seem to match what is shown in Figs. 5 or S11?

These were incorrect and have been updated (line 275).

27. Lines 216-218: Perhaps it would make sense to add references to recent analyses quantifying rates of compositional change (here or elsewhere in your discussion), e.g. Blowes et al. 2019 (as before), Pilotto et al. 2020, Burrows et al. 2019.

Pilotto, F. et al. Meta-analysis of multidecadal biodiversity trends in Europe. Nat. Commun. 11, 3486 (2020).

Burrows, M.T. et al. Ocean community warming responses explained by thermal affinities and temperature gradients. Nat. Clim. Chang. 9, 959–963 (2019).

Agreed, citations added (line 284).

28. *Line 223: Again, I think the reference to Vellend et al. 2013 here might be misused, as they did not evaluate compositional change, correct? And again I would suggest adding Blowes et al. 2019 here, as it reinforces and expands on the Dornelas et al. 2014 study.*

This sentence is intended to group together various results, some of which show no change in richness (Vellend et al. 2013), some show the same and contrast it with compositional change (Dornelas et al. 2014, Gotelli et al. 2017), while others address community level metrics like the distribution of biomass/abundance within the assemblage (Jones et al. 2018). The use of the singular noun “A recent landmark analysis” was a grammatical error and has been replaced with “**Recent landmark analyses** of temporal patterns in biodiversity **have** detected no systematic change in species richness or structure in natural communities, despite rates of compositional turnover greater than predicted by stochastic null models” (line 288-290).

29. *Line 233: What is a “macroclimatic niche”?*

Here we have replaced ‘macroclimatic’ with ‘climatic’. Essentially we are referring to the fundamental niche defined in environment space (line 327, 331).

30. *Lines 232-237: Would these patterns also be influenced by dispersal ability additionally to species having wide niche space/large ranges?*

According to our analysis of autonomous turnover, it requires that species distributions represent a subset of their fundamental spatial niche, i.e. it requires the existence of unoccupied patches into which the species could invade and effectively compete with local residents. Thus it follows that indeed for high dispersing and/or weakly interacting taxa for which all or most of the available habitat can be readily occupied, we would not predict the strong autonomous colonisation-extinction dynamics we report here. This distinction can be seen in the scan of the parameter space in Figure S3. For clarity we have replaced this sentence with the following: “The precise location of the transition between [fixed-points and autonomous turnover] depends on details such as dispersal traits, the ecological interaction network and environmental gradients (Fig. S3)” (line 318-320).

31. *Lines 239-241: Could the Authors mention something about how temporal resolution in their simulated communities compare to that of empirical data? Would it be relevant for seasonal, yearly, decadal analyses? For instance, the data included in the BioTIME database is at a yearly resolution, and I would argue there is a lot of data in this source and elsewhere too (not trying to make a “nasty comment”, just wondering how the simulations time “range” and “resolution” would match to empirical data 🙄)*

Again see our response to point 1 above.

32. *Lines 251-254: Really nice final sentence.*

Figures

33. *Figure 1: Does the figure correspond to “large model metacommunities” (there is also small N)? Doesn’t the figure describe the “Development of autonomous turnover as metacommunities size increase”?*

We have updated the title of this figure since the reviewer is correct to point out that not all metacommunities shown are “large”.

Does the figure show BC similarity or dissimilarity? Please indicate here and/or in the text what the range 0-1 means for the similarity (or dissimilarity?) metrics.

This is BC similarity, hence the 1.0s on the diagonal and this has been noted in the caption for the figure. Due to the length of the time series these figures are indeed rather challenging to interpret. We considered a narrower time window or coarser resolution, however in the former case, it is not clear that composition can return to similar states after long gaps at intermediate N, nor that little or no slowdown in the dynamics is evident, while in the latter case, the sharp transitions characteristic of the Clementsian turnover phase are harder to see.

34. Figure 2: I think this Figure could be in the Supp. information; it was not super clear to me the difference between the blue and red trajectories in the actual figure without reading the legend.

We consider this figure to be very useful in communicating the idea of a heteroclinic network so have left it in the main text. It also helps to clarify the qualitative difference between the cases with and without propagule pressure, with the latter undergoing dynamical aging - a progressive slowing of the dynamics - which is not the case for the former. We have added a legend to the figure to drive home the distinction between trajectories with and without propagule pressure, which should hopefully help clarify.

35. Figure 4: What is " α_{src} "? Also, perhaps need to define/explain spatial turnover here.

α_{src} is the local species richness of populations capable of persistence in the absence of immigration, i.e. source populations. This is a useful metric since it is largely independent of the spatial/abiotic parameterization. In contrast, measures of local species richness defined by assuming an arbitrary detection threshold are not only dependent on the chosen value, but more importantly do not strictly correspond to the intrinsically regulated component of the assemblage.

Here we use the mean BC *dissimilarity* to indicate the regionalization of the metacommunity that occurs during assembly which continues after average local species richness reaches local limits. While spatial beta diversity does not strictly contribute to our analysis of the emergence of autonomous turnover, we feel that including both spatial and temporal metrics in this figure provides important context.

Material and Methods

36. Lines 384-385: Suggest adding this detail to main text.

While we have not explicitly stated the covariance function for the random field in the main text, it is verbally described in the updated results section. The new Figures 1 will help to clarify this point further.

37. Lines 390-391: Does this mean the patches are constantly losing biomass? Why is this implemented/necessary, and how does this relate to populations/communities reaching the equilibrium states?

Indeed that is true. In practise however the emigration rate can be removed without loss of generality by simply absorbing it into the growth rate matrix. The inclusion of the negative

diagonal in the matrix **D** can be helpful to some readers since some biomass losses are expected as a result of dispersal away from a patch.

38. Line 410: “domain decomposition of the spatial network” – what does this mean?

“Domain decomposition” is a standard method to speed up simulation of spatially extended systems using multiple CPUs (parallelization). We feel that the rather involved computational methods employed in this technique are beyond the scope of this paper and instead will form a stand-alone publication in future. In order to clarify what is meant by ‘domain decomposition’ in this context we have added the following sentence:

“... we developed a parallel implementation of the assembly model that makes use of the algorithmic domain decomposition method (Gander 2001) for the population-dynamical simulations. This involves decomposing the metacommunity into spatial subdomains of equal numbers of patches, each of which is simulated by a unique parallel process (CPU), with boundary states regularly broadcast between processes” (line 577-582).

39. Line 441: Please specify what δA is, and consider if this can be confused with the competition coefficients notation?

We have changed some of the notation here and given more details on the method for generating the spatio-temporal subset of the metacommunity. We have retained the notation δA , however, but feel that the updated paragraph leaves little room for confusion with the interaction matrix **A** (line 610-623).

Supplementary text

40. Line 454: So “variability of environment” did not vary?

No, σ^2 and l were set in the same range and in all combinations. We are now using set notation ($\in \{\dots\}$) for greater clarity (line 633-634).

41. Lines 463-468: This is obviously ideal for implementing the simulations with the purpose of detecting autonomous turnover, but then begs the question of how “specific” or “general” are your results right?

This is of course a potential issue that cannot easily be resolved using numerical methods. However, we find little evidence to suggest that the phenomenology we describe depends on this specific combination of parameters, not least because the same basic patterns emerge in an isolated Lotka-Volterra model. Also note that, while we choose to explore the space of N while keeping the rest of the parameters fixed, autonomous turnover is observed throughout much of the space of ϕ , σ and l . We have added a discussion of this to the text (lines 132-135).

And related, is there any sense of how this parameter space is likely to match up/represent empirical communities?

Current ongoing research is exploring which parts of the parameter space are commonly represented in empirical communities. Indirect evidence that we are in the right ballpark comes from our comparison of empirical and simulated biodiversity patterns in Fig. 6. At this point we cannot directly support the biological plausibility of the chosen parameters with published evidence, however this is implied by the current research phase.

42. Lines 471-474: *The multiple-site dissimilarity metrics developed by Baselga are affected by the number of sites involved in the calculation, and this should be controlled for, as the effects on dissimilarity would be driven by the number of sites included (see Baselga 2010). Can the Authors please clarify and check if necessary when using the multi-site metrics (assuming that comparing extents with different number of sites would be similar to comparing neighbourhoods with different number of nodes).*

Thank you for pointing this out. In practice, since the size of the neighbourhood is 1 plus the degree of the focal node, and the degree in a Gabriel graph ranges from 1 to 8 with the majority of nodes falling in the range 3-5, the bias will be small. Furthermore, since we are averaging over all nodes in a given metacommunity, it is likely that the *average local neighbourhood spatial beta diversity* will be largely unimpacted by small deviations in the number of patches considered. However, we have now changed the analysis, using instead a metric devised by Legendre and De Caceres (2013) which normalises by the number of sites/time points. This is then applied to both the spatial subset *and* the time series for greater consistency. We note that the outcome is almost identical, but appreciate the reviewer making us aware of this issue. See updated figure S10, and lines 650-652.

43. Line 586: *Missing references?*

Again we have added reference to work on the portfolio effect (line 762).

44. Figure S8: *Specify if Bray-Curtis is similarity or dissimilarity; missing figure legend for heat-maps.*

We draw the reviewer's attention to the caption figure S8 which does specify that these are BC similarity matrices. However, the absence of a colour scale is indeed an omission which we have corrected.

45. Figure S9: *Specify if Bray-Curtis is similarity or dissimilarity.*

See our reply to 42 above. The metric we are now using takes the Bray Curtis *dissimilarity* matrix as an input and the output is maximized at 0.5 (high compositional dissimilarity).

Reviewer #2 (Remarks to the Author):

The manuscript by O'Sullivan and colleagues seeks to test whether autonomous, or competition-driven, turnover of species through time can reproduce observed patterns of temporal change in species diversity. The study finds that simple Lotka-Volterra competition models embedded in a metacommunity framework can indeed produce substantial autonomous turnover of species through time once they are made sufficiently large (in their number of local communities) to overcome species' extinctions and community collapse. Rates of temporal turnover of species well match those in empirical studies and spatial scaling of species richness across area and through time are also can match well those of empirical studies.

Overall, I do not find the results too surprising. We know that models with pure stochastic drift and neutral community dynamics (e.g. Hubbell's neutral model) that is more rapid than natural rates of speciation for communities under most circumstances (e.g. except when

dispersal is very limited). It is also not surprising to me that a metacommunity model can be tuned to match empirical patterns. In part because the empirical patterns do not contain a large amount of information and could be produced by many possible mechanisms, so a match of patterns at the level conducted seems like weak evidence for the importance of autonomous turnover.

We must respectfully disagree with the argument that our results are not surprising and believe that the reviewer's case to this effect is not in fact well supported. Neutral drift driven by demographic stochasticity (as it occurs in Hubbell's neutral model) has been excluded as a potential explanation for observed turnover in previous work (see Dornelas et al. 2014). Neutral drift is simply too slow to explain the observed rates of turnover. Furthermore, the reviewer does not provide evidence to support the statement that the empirical patterns that our model matches are easily matched by (meta)community models. In fact, there is strong evidence to the contrary. As an example, in their recent monograph on *Metacommunity Ecology*, Leibold and Chase devote the entire Chapter 5 to emergent spatio-temporal patterns, but do not cite a single model that would reproduce any of the patterns that emerge in the present study, let alone all of them together! Nor have we seen any of such discussion in the recent literature. If there are other studies which report similar patterns using different methods we would appreciate if the reviewer could point these out. According to our reading of the scientific literature, however, such ecological pattern formation has yet to be meaningfully described using theoretical models.

Studies such as reference 1 suggest that richness is anchored by something, and the typical explanation is niches. Are the results of the present manuscript not indicating that if we tune a manuscript with appropriate R values, competition coefficients and dispersal rates (and the consequent competition-colonization tradeoffs that are set or emerge in surviving species) we can achieve something like this? I think we have known this for quite some time.

Unfortunately, we are not entirely clear on what is meant by "this" in the last two sentences of the reviewer comment. We interpret this statement as referring to "niches" and "richness being anchored by something". But neither does the fact that species richness is regulated in the type of assembly models that we study depend on tuning of parameters (regulation of richness is observed in any population-dynamical community model with stepwise assembly that we are aware of, and there is strong mathematical theory to explain why), nor is regulation of species richness the central topic of our manuscript. Rather, the present study is concerned with secondary patterns that emerge in ecological systems subject to such intrinsic regulation.

For real communities our difficulty is to distinguish environmental covariation of species from actual interactions: negative interspecific covariation in responses to environmental conditions produces patterns that look like interspecific competition. It is also hard to quantify competition for more than a dozen species and even that number creates a large number of combinations of species that could interact.

We entirely agree with the statements in this passage and would argue that this is precisely why ecological models which employ randomness as a substitute for such unmeasurable biological quantities have been so influential in the development of the theory of community ecology. The extent to which this statement can be seen as an argument for or against the quality of our manuscript, is not clear to us. That such difficulties exist is clear; our study circumvents them in an elegant way.

The arguments in the two paragraphs of lines 26-38 that we can show that autonomous species interactions could be responsible for temporal change in communities seem overly simplistic. We fully expect that there are environmental drivers of change in communities: a large literature on disturbance supports this. Also we expect that species interactions can drive dynamics, a large literature on species interactions and ecological succession support this. It is not clear to me how the knowledge gained from the current work would help an ecologist determine that environmental drivers are unimportant or that species interactions are.

The salient point that the reviewer might have missed here is that, unlike the case of succession processes, the dynamics we observe are persistent. Since this is fundamental to the importance of our study we have added a paragraph in the introductory section explaining more clearly the concepts of autonomous dynamics and autonomous turnover that are at the centre of our work (see also our replies to reviewer 1 above). They relate to dynamics in general, and community turnover in particular, that do not cease even though there are no external drivers of those dynamics. Successional dynamics are not of the autonomous type according to this definition. Instead they require a particular starting point (the bare habitat), frequently involve biotically mediated alteration of local environmental conditions (ecological engineering) and, in simple models, come to a halt when the climax community is reached. The autonomous turnover we report here is not equivalent to that described by the theory of ecological succession, nor would it be interpreted as such in empirical studies of such a phenomenon.

The metacommunity model is described to have local variation in R , so that spatial variation in the environment is included but the local values of R are presumably fixed through time. Species dispersing across space creates a move from $R_{i,x1}$ to $R_{i,x2}$ so the species change their R -values through dispersal. Consequently, the amount of variation in R -values seems important to the idea that species turnover is purely autonomous. This at least needs a clearer explanation if not more work to determine its importance.

Here the reviewer speculates on what the mechanism driving autonomous turnover in our model might be, however we note that this appears to be somewhat different from the explanation offered in the first paragraph (“[invasion from the regional species pool] overcome species’ extinctions and community collapse”). In both cases, however, the reviewer does not refer to the long passage in our manuscript in which we explain and demonstrate what the true mechanism driving dynamics in our model is. Our explanation differs from both of these ideas: turnover is driven by invasions that lead to deterministic extinctions, extinctions that then permit new invasions. If the reviewer remains unconvinced by our arguments we respectfully request that they point out the erroneous steps in our reasoning rather than speculating on alternative hypotheses for which we have found little evidence. We believe that our examination of the mechanism that leads to autonomous turnover is robust and, in the absence of a clear counter-argument, the reviewer’s request in the final sentence is unclear to us.

Reviewer #3 (Remarks to the Author):

This is an intriguing work that tries to show us only species interaction and dispersal can exhibit turnover pattern observed in nature, providing us an alternative explanation other than environmental change and stochasticity. This work presented interesting results and mechanistic explanations. The results of the meta-communities and isolated LV systems

were well matched. However, there are still several aspects I feel the authors might need to work on. 1) The introduction can be re-edited to make it more smooth and trackable, mainly introducing the patterns the authors want to test in the results. 2) The methods need more details about the simulation. 3) some results need more clarification. My specific comments are listed below.

A major concern is the writing style of Introduction. I found it difficult to follow the flow of MS. Especially confused by what it is expected to see in the results. Therefore, in the introduction, maybe introduce what the readers expect to see about the turnover patterns observed in real systems and also list those they want to compare with their simulation.

Following comments from both Reviewers 1 and 3 we have now completely overhauled the introduction. Please see our responses to Reviewer 1 above for more details. Note that in the updated introduction we now refer directly to species-time-area-relationships (line 37), however we delay further discussion of the phenomenology of this pattern until the corresponding section on the macroecology of autonomous turnover.

Fig. 1B, it is the pattern in a local community. Do all local communities have a similar pattern? Is it a typical one? Or they are showing a specific case?

In Fig. 2B (previously Fig. 1B) we are showing temporal turnover in single nodes, since this is the most relevant scale both in terms of the mechanism we explore later in the manuscript, but also in terms of empirical comparison. In each simulated metacommunity turnover rates vary over the landscape. As seen in the colour gradient in Fig 2A, for metacommunities of $N \leq 32$, there are cases of local communities that are compositionally static as measured by hierarchical clustering (blue coloured nodes; note that since cluster analysis requires the application of thresholding, these can in fact include local communities with weak steady state population dynamics). For $N \leq 32$ we have selected nodes that do indeed manifest compositional turnover and indicated the chosen node in Fig. 1A. For $N > 32$ the BC similarity matrices are representative of all or most of the nodes in a given metacommunity simulation.

My concern is what the authors wanted to show for reader, the temporal pattern in single local communities or meta-communities?

Turnover occurs arguably at both scales in that populations move around in space (<https://vimeo.com/379033867>), however quantifying this at the regional scale is methodological challenge and contrary to standard empirical methods. Furthermore, in our discussion of the mechanism responsible for autonomous turnover, we show that in fact the relevant spatial scale is indeed the local scale: it is the onset of structural instability at the local scale and the resultant emergence of stable heteroclinic networks in community state space that best explains the temporal dynamics. Therefore the local scale is the focus of Fig 2.

Is the transition of turnover pattern prevalent in empirical meta-community? I am asking this because the authors declare the simulated patterns are observed in natural systems.

While there is currently little evidence of such a transition (but see Martin-Platero et al. 2018), this does not mean that the patterns we observed, or the mechanism responsible, are not active in natural ecosystems. It is very possible that current biomonitoring technologies do not have the temporal resolution required to detect this transition.

Martin-Platero et al (2018) for example observed clear compositional clustering among plankton communities using very high resolution time series. It's also possible that selection pressures absent from our metacommunity models preclude the long-term persistence of assemblages undergoing Clementsian temporal turnover. This would not, however, imply that the autonomous process we report is not active in natural ecosystems.

As described above, we have added a paragraph to the discussion about the order of magnitude of the rate of compositional change predicted by our model, and offer a comparison with empirical observations.

What does the color gradient mean in Fig. 1C?

This colour gradient visually underlines the observed qualitative change in metacommunity dynamics, quantified by the data points. Furthermore, the colours match the gradient used in the phase portrait in Fig. S6. The intention is to highlight the fact that the discrete transition between the UFP (orange) and what has been termed the MA (green) phase predicted by Bunin (2017) is an incomplete picture of the actual temporal dynamics, which in fact includes a gradual transition between oscillatory, Clementsian and Gleasonian turnover, represented by the saturation of the green phase of both figures.

Which part of text corresponds to Fig. 1C?

The reviewer is correct to point out that the third panel in figure 2 was not properly discussed in the text. We have added the following sentence to the results section which we hope clarifies its interpretation:

“The number of compositional clusters detected typically varies within a given metacommunity (Fig. 2A node colour), however we find a clear increase in the number of local clusters in composition, i.e. an increase in the rate of compositional change for time series of fixed length, with increases in the total metacommunity size (Fig. 2C)” (line 153-155).

L186: Temporal occupancy is TO_{ix} . What is the data for the density distribution in Fig. 5A,E. Is it TO_i averaged over all sites or TO_x averaged over local species or TO_{ix} ?

In Fig 5A this distribution was generated by computing the normalised temporal occupancy for all species in all nodes for all ten replicates assembled for the reported combination of parameters. These were then pooled into a single dataset so these are TO_{ix} at the metacommunity scale. It is likely that the temporal occupancy distribution for single nodes differs from that observed after pooling (again we refer to the differences in turnover rates within metacommunities shown in Fig 2A), however this method corresponds to that employed by Coyle et al. (2013), which we cite, who pooled temporal occupancy data for 492 distinct survey routes of the North American Breeding Bird Survey.

L 214: Fig. 5. All of a sudden, the readers see the comparison between bird and simulated community, then fish data were presented. Why different dataset?

Of the three empirical data sets used for comparison in Fig 5 each reported only one of the temporal occupancy distribution, the temporal species richness, or the exponents of the STAR. Empirical data-sets are usually not sufficiently complete to extract the full complement of known biodiversity patterns. We would argue that the fact that three

different patterns reported in three different empirical studies emerge in our models is more convincing for pattern-orientated model validation than if we would compare model outputs to a single, limited dataset.

First, it is better to introduce them in introduction. Second, one can often find a specific dataset with similar pattern to a model; my question is how a single example can prove that the pattern observed in model community is commonly observed in natural communities? or the authors just want to show a similar empirical pattern as in theoretical model?

We feel that, while it would be helpful to introduce the STAR and extensions early in the manuscript, this would divert attention away from the rather complex theoretical ideas that need to take precedence in the opening section. We do briefly refer to such patterns in the updated introduction (line 37) but withhold specific details until later in the manuscript.

Scientific theories are rarely “proven” by empirical data. But they can be disproven. A trustworthy theory is one that has been exposed to many comparisons with empirical data without displaying a fundamental inconsistency. We are aware of plenty of interesting ecological simulation studies that largely avoid comparison of patterns in simulation outputs with observed patterns. Yet, we believe that, if a theoretical study does include such comparisons with empirical patterns *and* reveals good agreement, then there is no question that this strengthens the trustworthiness of this study. Incidentally, we did not pick the empirical patterns to compare against for their good agreement with the model. We picked them because they were known and known to be empirically robust. Then, to our own surprise, we found that our model does reproduce them. We believe that this is how science is supposed to be done.

To further strengthen the comparison of our model with data, we note the new paragraph in the discussion in which we argue that temporal turnover rates are roughly consistent observations.

We do not claim that our results categorically prove or disprove that autonomous population dynamics of the kind we report here contributes to observed turnover. Above all, what our model highlights is that the assumption that autonomous dynamics are unlikely to sufficiently explain observed patterns in temporal turnover is not scientifically well supported. And we provide a stepping-off point for further work on quantitative analyses of the intrinsic drivers of temporal beta diversity.

L214: Fig. 5CD, are the data the same with ref.39? maybe mention that it is redrawn from ref.39.

Fig 5C and D are both redrawn from ref 14 (Adler et al. 2005). This is mentioned in the caption.

L228: The theoretical model shows empirical turnover pattern might emerge from just species interaction and dispersal. But I find it uneasy to apply it as null model because the parameters of the theoretical model like competition coefficient, dispersal rates are not easy to get. Otherwise, one has to randomly draw parameters, but this adds stochasticity to the model.

This is of course true. But it is not our fault. Sometimes nature is just not that simple. And this fact should not necessarily prevent science from progressing.

For example, important concepts of solid state theory, such as band-structure, the distinction between electric “conductors”, “semi-conductors” and “insulators” based on this band-structure concept, and the theoretical understanding of the consequence of doping semiconductors, which then led to the construction of the first solid-state transistors, all preceded the construction of the computers that were used to later accurately solve the quantum-mechanical equations from which these conceptual distinctions emerge. Had physicists dismissed the early conceptual advances because some aspects of solid state physics could not be predicted quantitatively at the time, then there would be no smart phones today.

We agree that more research is needed before we are able to empirically distinguish autonomous from externally driven compositional change. Work on fitting the LVMCM to empirical data is ongoing and may provide the foundation for a quantitative null model in future. Similarly, further developments of the analytic theory of metacommunities stimulated by this study may offer more robust predictions of baseline autonomous turnover.

Technique points:

The methods section needs more details.

L63: “spatially correlated normal distribution”. Could they explain more about ϕ and σ_2 , give the formulation? Or give a reference or package for this.

We have substantially updated the introduction of the manuscript, including two additional figures (see above). We hope that this will clarify concerns regarding the technical details of the model, and allow the reader to navigate the main text without first reading the Methods section in which further details are provided.

L377: pls give the integrator for the equation S1 somewhere in the methods

We are not entirely sure what is being requested here, but believe the question refers to the ODE solver that we used. We have therefore added the following sentence to Supplementary Information:

“After invaders were introduced, metacommunity dynamics were simulated using the SUNDIALS (Hindmarsh et al., 2005) numerical ODE solver” (line 568-569).

L398-408: The assembly of meta-communities needs more clarification and details. I am not sure if I get it right. In each simulation, did they invade the meta-community with $0.05S + 1$ first, then numerically solve the equation S1 from t to $t+1$, then invade again, until $t=1e+4$. Then use the meta-community data from $t=0$ to $t=1e+4$ to analyze temporal beta diversity? Or species invade, then solve S1 from $t=1$ to $t=1e+4$, then invasion, solve S1 from $t=1$ to $t=1e+4$, repeat it until intrinsic limit on the number of species regionally. Or apply the assembly process first, then numerically solve the equation S1 from $t=1$ to $t=1e+4$?

We concede that our description of the assembly algorithm was too light on details. We have therefore added the new figure S1 which we hope clarifies the process. In short, we add $0.05S+1$ species, then simulate for 500 unit times, remove all extinct species, and

repeat until the community saturates. Then switch off all external perturbations (invader flux) and study the steady state dynamics after allowing the system to relax for a long time (10^4 unit times). Please see also our related replies to Reviewer 1.

Please explain the simulation in detail, because this is the core method. A workflow plot or pseudo codes will be more clear. Since the codes are not provided, it is crucial for readers to know exactly how the authors simulated the meta-community dynamics.

Again, on this front please see the new Figures 1 and S1 which we hope will make the assembly algorithm and the various components of the model clearer.

L400: "The invaders were tested to ensure positive growth"? How do they test it? The invading species was introduced to all local sites, or a random site?

In order to clarify the testing process we have added the following sentences to the Materials and Methods section:

"The invaders were tested to ensure positive growth rates at low abundance. This was done by introducing a multiple of $0.05S + 1$ newly generated species into all patches at very low abundance, then simulating for a handful of time steps and testing for increasing biomass trajectories in at least one patch. Of the successful invaders, $0.05S+1$ were randomly selected and each introduced at 10^{-6} biomass units into the patch in which its growth rate was greatest during testing" (line 563-568).

L427: I find the terminology is sometimes confusing. In methods, it is termed spatio-temporal compositional similarity. It seems to me a novel measure, but just indicates spatial/temporal similarity.

Agree, this choice of terminology was an error, spatial/temporal is a more appropriate usage.

L402: "The metacommunity was periodically scanned", what is the period/time interval.

To clarify, the period of both the invasion and extinction processes described in figure S1 was 500 unit times. We have made the following clarification:

"The time between invasions we kept fixed at 500 unit times, and before each new invasion the metacommunity was scanned and species with biomass smaller than 10^{-4} biomass units in all nodes of the network were considered regionally extinct and removed from the model" (line 569-572).

L469, and Figure S9: Pls explain this "the time averaged spatial community dissimilarity at the local neighbourhood scale" in detail.

This refers to the spatial dissimilarity of the local neighbourhood averaged over the time series. Note that though the local neighbourhood composition turns over autonomously in time, the neighbourhood dissimilarity is largely time invariant and as such the local neighbourhood dissimilarity at a single time-point would be sufficient for this analysis.

What is the link between "the time averaged spatial community dissimilarity at the local neighbourhood scale" and the three parameters? What does each data point in Fig. S9

represent? How can a community dissimilarity index represent the three environmental parameters?

In combination, these three parameters determine the compositional dissimilarity of adjacent nodes by varying the between-patch differences in environment and the intensity of propagule pressure and resultant mass effects. Although environmental heterogeneity and dispersal traits impact biodiversity via fundamentally distinct mechanisms, we can distill the biotic response to these various processes into a single response, the spatial beta diversity. Since our analysis reveals that the most relevant scale for autonomous compositional change is in fact the local neighbourhood, we simplify the parameter space by computing the local neighbourhood spatial beta diversity. Our previous wording of the approach was not very clear. We therefore reworded this passage (lines 646-654).

Note that, following the advice of Reviewer 1, the metric used to compute the spatial/temporal beta diversity for this analysis is now that derived by Legendre and De Caceres (2013).

Reviewer comments, second round

Reviewer #1 (Remarks to the Author):

I thank the Authors for their detailed responses and for the extensive changes made to the manuscript and the additional figures. I think these go a long way to improve the clarity of the manuscript and of what the developed models can do; also, re-focusing the introduction and discussion more to the ecological issues will likely improve the impact of the work presented. I think the paper reads very well, and will be an important contribution. I do not have any major suggestions, and provide only a few additional comments and suggestions, which I hope the Authors will find useful.

1. Perhaps you could add to the Abstract a short sentence to indicate the mechanisms/processes that yield the autonomous turnover in your models (if not going over the word limit).
2. While the new Introduction is much clearer, it is perhaps now missing the general point that turnover can come about from environmental change, drift or other processes, so perhaps you can re-up the sentence(s) you had in the first paragraph of your previous version to the beginning of the Intro (i.e. before describing autonomous turnover). I think the same is true for the small paragraph in lines 33-38 in the previous version, now deleted – this bit might be useful for setting up why developing models like yours is important, and also highlight the potential implications in real life, e.g. for conservation/management.
3. Line 38: Not clear what is meant by “underlying consistency accessible through modelling”.
4. Lines 39-40: This sentence seems very similar to the one in lines 20-21, but now referring to antagonistic interactions instead of competitive networks.
5. Lines 45-46: This sentence is hard to follow aside from a pure methodological perspective, and it seems circular... What is meant by “characteristic spatial structures”? And both characteristic and spatial are misspelled.
6. Also consider if adding “directional” would clarify what is meant by “Acyclic turnover” in this paragraph.
7. Line 49: Should “population turnover” be “community turnover”?
8. Line 53: Delete “Here”.
9. Lines 83-95: In my opinion, this paragraph seems to be out of place and cuts the flow between the first introduction of the metacommunities and how they are established in your simulations; consider moving somewhere else?
10. Line 91: Please add these were shown in simulated metacommunities.
11. Line 127: “full characterisation *of* autonomous turnover...”.
12. Line 138: Isn't 8 the minimum N?
13. Line 152: Not sure the use of “unpredictable” is correct here, i.e. was compositional change predictable at the smaller scales?
14. Line 162: Which mechanism is this?
15. Line 184: “appears to remain unclear” - weird wording. Can you be a bit clearer how this is relevant for ecological turnover dynamics?
16. Line 193: Perhaps add here a “so that...” type of sentence to help the reader follow these steps.
17. Line 214: “Heterogeneities” rather than “inhomogeneities”?
18. Lines 263-265: As I mentioned in the previous version, and after reading the Authors response to my comment, I still think this statement is too strong, insofar as it rules out any other potential explanations. What the Authors state may be the case, but we cannot be certain that autonomous turnover is the only driver, since rates of turnover can be more or less constant (with species coming in and going out periodically, even in the presence of environmental changes), and there can also be simple drift or core-transient dynamics e.g. – the point being that in all these scenarios we could observe relative stasis in species richness and abundance, and the changes in species composition may not directly result from environmental forcing. My point here is simply that there may be other mechanisms at play and that stating that if richness is stable in the absence of direct environmental change, any turnover is autonomous is too peremptory.
19. I think you could add to your discussion a few general, quite short statements about the utility and development of the models, as per your response letter (e.g. potential developments to

quantify such autonomous turnover with a view to build null models). But more importantly perhaps, regarding the effect of the particular combination of parameters used to generate the metacommunities analysed, you have indeed added a sentence about exploring the parameter space, but looking at Fig S3 and the associated supplementary text, it is clear that different combinations could lead to different results, while the variation in these parameters likely represents different types of empirical metacommunities, which would warrant further exploration.

20. You mention in your response that the selection of the competitive coefficients were based on an arbitrary choice. Is there any sensitivity analysis on this decision? And do you have any sense whether those values can somehow be representative (at least comparing to some of the few empirical studies that might have been able to estimate such competition coefficients)? Only a comment, not necessarily something to affect the ms.

21. There is a switch between similarity and dissimilarity in the text, and being shown in different figures. Consider if it would be better to have consistent metrics throughout (if possible, or make sure every instance is clearly indicate which metric is used). One example is you use "dissimilarity" in line 591 but then refer to "BC similarity" in line 601. Please add which metric was used to Fig. S10 legend.

22. Related to this, consider if the result shown in Fig S10 and described in lines 652-668 could be mentioned in the main text, particularly in lines 666-668.

23. This paper by Lewandowska et al. (<https://onlinelibrary.wiley.com/doi/full/10.1111/geb.13078>) might be relevant for some of the points you make.

With my best wishes,
Laura Antão

Reviewer #2 (Remarks to the Author):

I am reviewing the revised version of the manuscript. I was Reviewer #2 for the first version of the manuscript during August 2020. I appreciate the efforts that the authors have gone to in revising the manuscript. I do find that the revised version is substantially clearer and more accessible to a broad audience. In particular, I appreciate the highlighting that rates of turnover from ecological drift are slower than those in natural communities (Dornelas et al. 2014). This information was present in the previous version of the manuscript but I failed to comprehend it and put it into the perspective of the manuscript's findings, which are now clear in the opening sentence of the main text. I apologize for this oversight and I am considerably more excited about the results now that I understand this context. I have some suggestions for clarifications and tidying up, and think it would be helpful to readers to give a bit more context for the extent to which the results do and don't match the empirical literature.

The manuscript by O'Sullivan and colleagues seeks to test whether autonomous, or competition-driven, turnover of species through time can reproduce observed patterns of temporal change in species diversity. The study finds that simple Lotka-Volterra competition models embedded in a metacommunity framework can indeed produce substantial autonomous turnover of species through time once they are made sufficiently large in their number of local communities to overcome species' extinctions and community collapse. Rates of temporal turnover of species well match those in empirical studies and spatial scaling of species richness across area and through time are also can match well those of empirical studies.

Lines 274-276. Can you more fully describe what the range of exponents were in the empirical data and what kinds of ecological communities they represent? Perhaps even plot a histogram of the exponents (or similar) from empirical data and show values from the simulations? As an ecologist I think this kind of evidence is critical.

Lines 322-325. I found the macroinvertebrates in the UK example strange. Many macroinvertebrates likely have extremely low ability to move and so to call these a metacommunity seems presumptive. The example seems to imply that all communities are

structured by metacommunity dynamics.

Figure 1. I like this figure in general but I would clarify it. The orange oval says "Autonomous population dynamics drive local compositional change." I found this confusing because I don't know whether it is talking about local or regional population dynamics. I initially interpreted as autonomous local population dynamics drive local compositional turnover, which leaves out immigration. If you say "Autonomous regional population dynamics drive local compositional change" this implies that dispersal is included, which is clearer. It would also be more direct to say dispersal rather than connectivity on the left side of the figure. Lastly, in the legend please spell out the element-wise multiplication used in the equation by defining the symbol.

Page 2 line 20. I suggest replacing "specific motifs" with what this represents since the language is not clear.

Page 3 lines 43-45. I'm not sure of the purpose of making an argument about cyclical vs. other forms of turnover. Since cyclical versus non-cyclical mechanisms are not clearly described the arguments are obscure. They are also poorly connected to the explanations on lines 174-193.

Page 3, line 49. The study is about compositional turnover (e.g. lines 27-28) not population turnover. I suggest to use consistent language.

Page 3, line 53. The emphasis is on local and regional compositional turnover through time, whereas spatio-temporal turnover implies joint consideration of turnover across space and time. Do you not mean just temporal turnover at local community and metacommunity scales? Also the emphasis of the manuscript is not on patterns, so I suggest to replace this word with composition or similar.

Page 4, line 73. "Between successive invasions" is unclear. In each simulation? A reviewer made a comment about invasions being a confusing word in the previous draft, and this holds in the current draft. I found the use of invasions on page 5 line 82 to be more clear, but even this would benefit from clarifying that it means the "introduction of new species;" non-native invaders often differ in their characteristics from native species, which makes the usage or intention potentially confusing. The term invader is then used again on line 172. Presumably some invaders are species (or close to them in parameter space to those) that have already been transiently present in a local community, which makes the term even more unclear when species identity is thought about.

Page 5 lines 84-88. The language here is inconsistent with the idea that there was a minimum (positive) threshold level of abundance in the simulations. Most simply, delete the sentence on lines 84-86 (it doesn't add much) and the parenthetical remark about non-negative abundances on lines 87-88. Simply saying something like the following achieves your aim? "Ecological structural stability is taken to describe the capacity of all species in a community to persist in the face of small biotic or abiotic perturbations." (Also note the typo on line 88 "aboitic".)

Page 5 line 92. Delete the word "strong." Indirect evidence is indirect evidence, so I'm not sure what is strong indirect evidence.

Lines 106-109. This sentence is too complex to easily read and would benefit from splitting it into two.

Line 127. Missing word "of".

Line 130. Clarify the wording. Allowed diversity to settle to regional saturation?

Line 135. Delete the unneeded word "reasonably".

Lines 153-155. Is this saying the number of compositional clusters per unit space also increases? Please clarify.

Line 162-165. I do not understand the context for these comments about large diverse communities having Gleasonian structure. Are they referring to the results of simulations or something that is in the literature (and unreferenced)? The paragraph ends abruptly and I was left feeling that I did not understand its purpose.

Lines 172-173. It seems convenient to draw on arguments about metacommunity size. However, size also alters environmental heterogeneity in your simulations. I'd suggest to make the arguments inclusive of this. Lines 212-214 are about this, but best to incorporate it from the beginning rather than it being added in part way through the section.

Line 278 correct "marcoecological." More generally, spell check the manuscript and supplement.

Figure S5 legend. Correct "communit".

Reviewer #3 (Remarks to the Author):

The authors have made substantial efforts to address most of my concerns. I agree that this work can stimulate the thinking about the internal forces of natural community turnover. The improvement on the writing is significant. A minor additional concern is the code availability for the simulation. In my opinion, for a simulation work, it is essential for others to replicate the results. Furthermore, the authors propose that autonomous turnover might be prevalent and distinguished from community turnover by external forces. For that, code availability and friendly usage will be important for more empirical ecologists to recognize this idea and apply it to future researches. The statement like "the code is available upon reasonable request" is no use because the authors can easily decline any request. I find no reason to hide the codes for a simulation study. Therefore, my suggestion for authors is providing codes with nice annotations on public repositories or appendix files for readers to replicate the shown patterns in this study.

REVIEWER COMMENTS and author replies

Please note that we have now updated the formatting of our manuscript to match the journal requirements for second submissions. For convenience we have included the PDF `main_unformatted.pdf` which is fully updated with respect to the second review but for which the formatting is the same as the previous submission.

Reviewer #1 (Remarks to the Author):

I thank the Authors for their detailed responses and for the extensive changes made to the manuscript and the additional figures. I think these go a long way to improve the clarity of the manuscript and of what the developed models can do; also, re-focusing the introduction and discussion more to the ecological issues will likely improve the impact of the work presented. I think the paper reads very well, and will be an important contribution. I do not have any major suggestions, and provide only a few additional comments and suggestions, which I hope the Authors will find useful.

1. Perhaps you could add to the Abstract a short sentence to indicate the mechanisms/processes that yield the autonomous turnover in your models (if not going over the word limit).

We have added the following sentence to the abstract:

'We find that autonomous turnover is triggered by the onset of ecological structural instability---the mechanism that also limits local biodiversity.' (lines 9-10)

2. While the new Introduction is much clearer, it is perhaps now missing the general point that turnover can come about from environmental change, drift or other processes, so perhaps you can re-up the sentence(s) you had in the first paragraph of your previous version to the beginning of the Intro (i.e. before describing autonomous turnover). I think the same is true for the small paragraph in lines 33-38 in the previous version, now deleted – this bit might be useful for setting up why developing models like yours is important, and also highlight the potential implications in real life, e.g. for conservation/management.

Responding to these suggestions, we have reinstated in the introduction the following text (adapted) from the initial submission:

'Potential drivers of such biotic change include changes in the abiotic environment, random demographic fluctuations (referred to as community drift) and autonomous population dynamics driven by ecological interactions and dispersal.'

In the second paragraph we have gathered ideas that had previous been somewhat diffusely expressed, relating to the potential interaction between autonomous turnover and conservation/management centred around the now explicit claim:

'The extent to which processes intrinsic to ecosystems contribute to turnover, however, remains poorly understood (Magurran *et al.* 2019). Understanding the expected amount of temporal

turnover due to such intrinsic processes is of vital importance if the causes of ecological change are to be accurately interpreted.’ (lines 26-29)

3. Line 38: Not clear what is meant by “underlying consistency accessible through modelling”.

We have updated this sentence to clarify the claim:

“Limited availability of historical turnover data before the onset of widespread anthropogenic impacts poses considerable challenges when trying to establish the natural baseline of turnover. Nevertheless, broad consistency amongst the species-time-area relationships observed in extant assemblages (Adler, 2005; White 2006) points to a common dominant biological process. It is reasonable to expect, therefore, that the drivers of spatio-temporal compositional turnover can be probed using theoretical models.” (lines 42-47)

4. Lines 39-40: This sentence seems very similar to the one in lines 20-21, but now referring to antagonistic interactions instead of competitive networks.

Indeed, this was not well structured. The paragraph including lines 20-21 was intended to explicitly define ‘autonomous’ processes. We used an example to help with this definition, however this is not strictly necessary. We have therefore moved the citation previously in line 20-21 to the paragraph previously including lines 39-40.

5. Lines 45-46: This sentence is hard to follow aside from a pure methodological perspective, and it seems circular... What is meant by “characteristic spatial structures”? And both characteristic and spatial are misspelled.

Apologies, we obviously missed a final spell-check. The “characteristic spatial structures” resulting from local oscillations include spirals and patch mosaic structure. Since the details of these structures are a side track, we only provide references. The key of the paragraph is the distinction between cyclic turnover (including chaotic cycles) of the type $A \rightarrow B \rightarrow \dots \rightarrow A$, and acyclic turnover which does not involve a return to ancestral compositional states. This distinction is relevant because acyclic turnover involves the injection of novel species, which, in nature, is a manifestly spatial process (arrival at a focal site from adjacent locales). While it would be premature to begin to describe the emergent patterns reported later in the manuscript at this stage, this paragraph is intended to highlight the necessity of the spatially structured metacommunity approach to studying temporal compositional turnover. In the absence of this argument, one might ask why the metacommunity framework is necessary for our analysis. We have edited the paragraph to make these points clearer. (lines 48-59)

6. Also consider if adding “directional” would clarify what is meant by “Acyclic turnover” in this paragraph.

Added.

7. Line 49: Should “population turnover” be “community turnover”?

Indeed. We have changed this to ‘compositional turnover’ to avoid repetition of ‘community’ in this sentence.

8. Line 53: Delete “Here”.

Done.

9. Lines 83-95: In my opinion, this paragraph seems to be out of place and cuts the flow between the first introduction of the metacommunities and how they are established in your simulations; consider moving somewhere else?

Agreed. This paragraph is central to our discussion since it defines the broader theoretical context within which our results are best explained. However, it was not best placed in the results section. We have moved to the introduction (lines 65-77).

10. Line 91: Please add these were shown in simulated metacommunities.

This sentence has been removed in responding to point 9 above to avoid repetition in the introduction. We now emphasize the simulations in metacommunities in the following new text (line 65). We hope that it is sufficiently clear in this sentence that the demonstration of the role of structural instability in spatially structured communities has so far been limited to ecological models.

11. Line 127: “full characterisation *of* autonomous turnover...”.

Added.

12. Line 138: Isn't 8 the minimum N?

It is true that $N=8$ is the smallest size studied here, but a combination of previous results and extrapolation from Fig 2C confirm that for smaller systems dynamics converge on fixed points.

13. Line 152: Not sure the use of “unpredictable” is correct here, i.e. was compositional change predictable at the smaller scales?

We have replaced ‘unpredictable’ with ‘acyclical’ which has previously be associated with ‘directional’ change in composition (line 160).

14. Line 162: Which mechanism is this?

The study cited here (Liataud et al. 2019) used a 1D metacommunity model with competitive Lotka-Volterra dynamics to show that biotic interactions can drive Clementsian turnover in space, with sharp transitions between compositional clusters along the single spatial dimension modelled. Related work on 0D communities (Bunin 2017) has described the parameter space in which we report the emergence of heteroclinic networks as a Multiple Attractor phase, which, we contend, is a miss-interpretation based on dynamical ageing typical of heteroclinics. It is plausible, therefore, that the Clementsian compositional gradients observed in the system studied by Liataud et al. (2019) are in fact driven by the same heteroclinic cycling we report here. It is likely that restricting analysis to 1D systems will greatly restrict immigration pressure and limit local neighbourhood diversity which could permit dynamical ageing that is avoided in much of the parameter space of the 2D model we describe here.

At this stage of our manuscript we are giving a purely phenomenological description of autonomous turnover via heteroclinic cycling. As such the arguments just made cannot be fully unpacked here. The citation is intended instead to stress that the view that competition can lead to surprising sharp transitions in composition and community clustering is an emerging hypothesis in theoretical ecology.

15. Line 184: “appears to remain unclear” - weird wording. Can you be a bit clearer how this is relevant for ecological turnover dynamics?

We have updated this sentence with the following, stronger phrasing: ‘This has been called the multiple attractor (MA) phase (Bunin, 2017). However, the implied notion that this part of the phase space is in fact characterised by multiple stable equilibria may be incorrect’ (lines 190-192). We hope this clarifies how this tangent is relevant to turnover dynamics.

16. Line 193: Perhaps add here a “so that...” type of sentence to help the reader follow these steps.

We have added a couple of additional clauses to this paragraph to try to ground it a bit more firmly in the idea of autonomous community turnover:

‘Population dynamical models with many species have been shown to easily exhibit attractors called stable heteroclinic networks (Hofbauer 1994), which are characterized by dynamics in which the system bounces around between several unstable equilibria, each corresponding to a different composition of the extant community, **implying indefinite, autonomous community turnover** (Fig. 3, red line). As these attractors are approached, models exhibit increasingly long intermittent phases of slow dynamics, which, when numerically simulated, can give the impression that the system eventually reaches one of several ‘stable’ equilibria, **suggesting that turnover comes to a halt.**’ (lines 193-200)

17. Line 214: “Heterogeneities” rather than “inhomogeneities”?

Updated.

18. Lines 263-265: As I mentioned in the previous version, and after reading the Authors response to my comment, I still think this statement is too strong, insofar as it rules out any other potential explanations. What the Authors state may be the case, but we cannot be certain that autonomous turnover is the only driver, since rates of turnover can be more or less constant (with species coming in and going out periodically, even in the presence of environmental changes), and there can also be simple drift or core-transient dynamics e.g. – the point being that in all these scenarios we could observe relative stasis in species richness and abundance, and the changes in species composition may not directly result from environmental forcing. My point here is simply that there may be other mechanisms at play and that stating that if richness is stable in the absence of direct environmental change, any turnover is autonomous is too peremptory.

This may indeed be too strong a claim. We have reworded as follows: ‘In the light of our results, we propose the absence of temporal change in community properties such as richness or the abundance distribution despite potentially large fluctuations in population abundances (Magurran and Henderson 2010) **as indicative of autonomous compositional turnover.**’ i.e. with the reference to a ‘predominant’ process removed (line 277).

19. I think you could add to your discussion a few general, quite short statements about the utility and development of the models, as per your response letter (e.g. potential developments to quantify such autonomous turnover with a view to build null models). But more importantly perhaps, regarding the effect of the particular combination of parameters used to generate the metacommunities analysed, you have indeed added a sentence about exploring the parameter space, but looking at Fig S3 and the associated supplementary text, it is clear that different combinations could lead to different results, while the variation in these parameters likely represents different types of empirical metacommunities, which would warrant further exploration.

The points you raise here correspond to on-going work, in particular regard to the fitting of the model to empirical systems by extracting abiotic and biotic parameters from data. At this stage we can only comment on the goals of this research, not its outcomes, but we have added the following sentences to the conclusion which highlights this potential application of the model:

“Future work will involve fitting the model described here to observations by estimating abiotic and biotic parameters from empirical datasets. In the supporting information we show how different combinations of parameters lead to different quantitative outcomes (S3), likely representing different types of empirical metacommunities. Understanding where in this parameter space natural systems exist may provide the foundation for a quantitative null model, a baseline expectation of turnover against which observations can be compared” (lines 312-318).

20. You mention in your response that the selection of the competitive coefficients were based on an arbitrary choice. Is there any sensitivity analysis on this decision? And do you have any sense whether those values can somehow be representative (at least comparing to some of the few empirical studies that might have been able to estimate such competition coefficients)? Only a comment, not necessarily something to affect the ms.

We have added a non-exhaustive but, we hope, sufficient sensitivity analysis to the supporting information which we believe addresses concerns about the arbitrary selection of a random distribution from which to sample the interspecific interaction coefficients. We did this by assembling model metacommunities with interaction coefficients sampled from two alternative fundamental distributions with the mean and variance of the off-diagonal elements the same as in Fig 5. We show that the outcomes are unimpacted by the choice of distribution.

While we do not explore this in the revised manuscript, we do see some quantitative differences in species richness when different values of connectance are used. Our analysis suggests that these occur due to the fact that, for a connectance of around 1.0, matching the mean and variance to that of the test case requires parameterizing the distributions in a way that permits extreme values such as $A_{ij} \sim 1$ or -1 . Such interactions are preferentially selected for or against during metacommunity assembly, altering the realised distribution of interaction coefficients and therefore the quantitative outcomes. This is a reminder of the fact there can be important differences between fundamental (sampled) and realised distributions of species interactions as a result of self-organisation, but it does not inform the present study.

A figure and the section ‘Distribution of interaction coefficients’ have been added to the supporting information summarising this sensitivity analysis (lines 613-650). We also refer to this section in the main text (lines 117-120)

Note that, for consistency with both the new Fig. S3 and the now public data and simulation software, we have updated Fig. 5 with a new simulation run. The outcome is the same, we have just used a fixed random seed so the result can be exactly reproduced.

21. There is a switch between similarity and dissimilarity in the text, and being shown in different figures. Consider if it would be better to have consistent metrics throughout (if possible, or make sure every instance is clearly indicate which metric is used). One example is you use “dissimilarity” in line 591 but then refer to “BC similarity” in line 601. Please add which metric was used to Fig. S10 legend.

We have now changed all uses of BC similarity to dissimilarity with the exception of figure S12 in which we show a decay of compositional similarity through time, which we consider a more intuitive representation than an increase of dissimilarity through time. In the caption for this figure we use italic text to emphasise the change of metric.

22. Related to this, consider if the result shown in Fig S10 and described in lines 652-668 could be mentioned in the main text, particularly in lines 666-668.

We discuss this result in the final paragraph of the section ‘Mechanistic explanation of autonomous turnover’, however with perhaps less succinctness than in the supporting information. We have added the following sentence to this section which is a reworking of the previous lines 666-668:

“Thus, we conclude that it is not species richness or spatial dissimilarity *per se* that best predict temporal turnover, but the size of the pool of species with positive invasion fitness, i.e. those not repelled by the combined effects of biotic and abiotic filters” (LINES 257-260)

23. This paper by Lewandowska et al. (<https://onlinelibrary.wiley.com/doi/full/10.1111/geb.13078>) might be relevant for some of the points you make.

Indeed, we have added reference to this paper to our introduction.

With my best wishes,

Laura Antão

Reviewer #2 (Remarks to the Author):

I am reviewing the revised version of the manuscript. I was Reviewer #2 for the first version of the manuscript during August 2020. I appreciate the efforts that the authors have gone to in revising the manuscript. I do find that the revised version is substantially clearer and more accessible to a broad audience. In particular, I appreciate the highlighting that rates of turnover from ecological drift are slower than those in natural communities (Dornelas et al. 2014). This information was present in the previous version of the manuscript but I failed to comprehend it and put it into the perspective of the manuscript's findings, which are now clear in the opening sentence of the main text. I apologize for this oversight and I am considerably more excited

about the results now that I understand this context. I have some suggestions for clarifications and tidying up, and think it would be helpful to readers to give a bit more context for the extent to which the results do and don't match the empirical literature.

The manuscript by O'Sullivan and colleagues seeks to test whether autonomous, or competition-driven, turnover of species through time can reproduce observed patterns of temporal change in species diversity. The study finds that simple Lotka-Volterra competition models embedded in a metacommunity framework can indeed produce substantial autonomous turnover of species through time once they are made sufficiently large in their number of local communities to overcome species' extinctions and community collapse. Rates of temporal turnover of species well match those in empirical studies and spatial scaling of species richness across area and through time are also can match well those of empirical studies.

Lines 274-276. Can you more fully describe what the range of exponents were in the empirical data and what kinds of ecological communities they represent?

One of the key conclusions from meta-analyses of the power law SAR and STR is that their exponents are remarkably consistent across location and taxonomic groups (White et al. 2006). We have added a sentence to this section describing the typical quantitative ranges observed, and the fact that these exponents show no strong signal of location or taxonomy (lines 290-293)

Perhaps even plot a histogram of the exponents (or similar) from empirical data and show values from the simulations? As an ecologist I think this kind of evidence is critical.

Unfortunately the original studies referred to here do not make their data available nor are they plotted in a way that can be readily extracted, either because they show only mean and sample standard deviations or because data points are too densely plotted to be amenable to reconstruction.

Lines 322-325. I found the macroinvertebrates in the UK example strange. Many macroinvertebrates likely have extremely low ability to move and so to call these a metacommunity seems presumptive. The example seems to imply that all communities are structured by metacommunity dynamics.

The rapid spread of invasive macroinvertebrates like *Dikerogammarus haemobaphes* (Guareschi et al 2020) throughout the UK suggests an important role for spatial processes in driving macroinvertebrate community composition. Similarly metagenetic monitoring of macroinvertebrates in Canadian wetlands (Bush et al. 2020) have been interpreted through the lens of metacommunity ecology. We would argue that the metacommunity framework has the potential to encompass a wide variety of ecological community types insofar as a smooth transition from single site/well mixed communities to spatially structured systems, those typically referred to a metacommunities, is likely to occur under variation of the parameters of any conceivable metacommunity model.

Figure 1. I like this figure in general but I would clarify it. The orange oval says "Autonomous population dynamics drive local compositional change." I found this confusing because I don't know whether it is talking about local or regional population dynamics. I initially interpreted as autonomous local population dynamics drive local compositional turnover, which leaves out immigration. If you say "Autonomous regional population dynamics drive local compositional change" this implies that dispersal is included, which is clearer. It would also be more direct to

say dispersal rather than connectivity on the left side of the figure. Lastly, in the legend please spell out the element-wise multiplication used in the equation by defining the symbol.

We have updated this figure and the caption in a way that we hope clarifies the distinction between the elements of the model (competitive interactions, a spatial network and some environmental heterogeneity) and its emergent properties. Autonomous turnover is not the only emergent property we observe but, as the focus of the paper, is the only one we refer to in the figure. While the time series shown in this figure could be described as population dynamics rather than turnover, they do show that species come and go at the local scale as a result of these autonomous fluctuations, so we hope the terminology is consistent. We have added a sentence defining the circle as denoting element-wise product.

Page 2 line 20. I suggest replacing "specific motifs" with what this represents since the language is not clear.

We have modified the introduction in response to reviews, hopefully now making it as clear as possible. This included removing the term 'motif'.

Page 3 lines 43-45. I'm not sure of the purpose of making an argument about cyclical vs. other forms of turnover. Since cyclical versus non-cyclical mechanisms are not clearly described the arguments are obscure. They are also poorly connected to the explanations on lines 174-193.

We have tried to clarify the relevance of cyclical vs acyclical turnover (line 54), the later being directional in nature and therefore considerably more realistic than the cyclical turnover that has previously been described in analyses of autonomously fluctuating dynamics in ecosystem models.

Page 3, line 49. The study is about compositional turnover (e.g. lines 27-28) not population turnover. I suggest to use consistent language.

Thank you for highlighting this sentence, we indeed intended to write 'compositional turnover' here.

Page 3, line 53. The emphasis is on local and regional compositional turnover through time, whereas spatio-temporal turnover implies joint consideration of turnover across space and time. Do you not mean just temporal turnover at local community and metacommunity scales? Also the emphasis of the manuscript is not on patterns, so I suggest to replace this word with composition or similar.

We say "spatio-temporal" in light of the interaction between spatial and temporal processes we observed in the STAR in models. It's true that we consider some patterns that are strictly temporal, such as the distribution of temporal occupancies. However, since the mechanism driving even those patterns for which a spatial dimension is not explicitly measured *is* fundamentally spatial – immigration from neighbouring sites and the resultant perturbation of the dynamical system – we feel the term 'spatio-temporal' is justified.

We feel that the term 'pattern' is also justified since we use it in the sense of 'Pattern-Orientated Modeling' (Grimm 2005) which invokes the exploration of the drivers of macroscopic structure in complex systems using models validated via qualitative pattern matching. In this sense the emphasis of our manuscript is indeed on patterns.

Page 4, line 73. "Between successive invasions" is unclear. In each simulation? A reviewer made a comment about invasions being a confusing word in the previous draft, and this holds in the current draft. I found the use of invasions on page 5 line 82 to be more clear, but even this would benefit from clarifying that it means the "introduction of new species;" non-native invaders often differ in their characteristics from native species, which makes the usage or intention potentially confusing. The term invader is then used again on line 172. Presumably some invaders are species (or close to them in parameter space to those) that have already been transiently present in a local community, which makes the term even more unclear when species identity is thought about.

It is true that our terminology in this regard lacked clarity. We have now replaced all references to 'inva[sion/der]' with 'regional inva[sion/der]' whenever this refers to the iterative metacommunity assembly algorithm. We also agree that the term 'invasive species' typically implies a particular set of characteristics relative to the resident community, however the term 'invasion' is not strictly reserved for discussion of 'invasive species' but can instead be used in other contexts ("invasion fitness", "local invasion", etc). It is this latter, broader notion that we employ.

Page 5 lines 84-88. The language here is inconsistent with the idea that there was a minimum (positive) threshold level of abundance in the simulations. Most simply, delete the sentence on lines 84-86 (it doesn't add much) and the parenthetical remark about non-negative abundances on lines 87-88. Simply saying something like the following achieves your aim? "Ecological structural stability is taken to describe the capacity of all species in a community to persist in the face of small biotic or abiotic perturbations." (Also note the typo on line 88 "abiotic".)

This is clearer, we have edited these sentences as advised.

Page 5 line 92. Delete the word "strong." Indirect evidence is indirect evidence, so I'm not sure what is strong indirect evidence.

Fair point. We do however wish to stress the potential import of the growing suite of empirically realistic macroecological patterns associated with structural instability since exploration of this phenomenon is a central research goal. We have replaced the word 'strong' with 'compelling' and hope we agree that indirect evidence can be impactful in this sense. Note paragraph moved (line 72)

Lines 106-109. This sentence is too complex to easily read and would benefit from splitting it into two.

Agreed.

Line 127. Missing word "of".

Corrected.

Line 130. Clarify the wording. Allowed diversity to settle to regional saturation?

We have replaced the phrase 'to regional saturation' with 'until regional diversity limits were reached'. (line 140)

Line 135. Delete the unneeded word "reasonably".

Deleted.

Lines 153-155. Is this saying the number of compositional clusters per unit space also increases? Please clarify.

These are clusters in time rather than in space, and determined at the scale of a single patch. To avert this misunderstanding we added "over time" in the relevant sentence. (line 162)

Line 162-165. I do not understand the context for these comments about large diverse communities having Gleasonian structure. Are they referring to the results of simulations or something that is in the literature (and unreferenced)? The paragraph ends abruptly and I was left feeling that I did not understand its purpose.

This is a modelling result. Large metacommunity models manifest Gleasonian temporal turnover. We have clarified that this result refers to models (line 172). The remaining sentence in this paragraph gives further qualitative description of Gleasonian temporal turnover.

Lines 172-173. It seems convenient to draw on arguments about metacommunity size. However, size also alters environmental heterogeneity in your simulations. I'd suggest to make the arguments inclusive of this. Lines 212-214 are about this, but best to incorporate it from the beginning rather than it being added in part way through the section.

In lines 172-3 (previous version) we discuss the 0D point model which is used to dissect the mechanism driving turnover in metacommunities. It's true that environmental heterogeneity enters the model in a rather different way in the 2D case; in the 0D analogue the ecological neighbourhood does not consider the fact that immigration rates in the full metacommunity model are directly impacted by the spatial autocorrelation of the environment. This simplification is explicit in the design of the 0D analogue.

Note that in the 2D case, heterogeneity in our model does not change with system size. Any square subsample of a large-area community contains (statistically) just as much environmental variation as a full small model community of the same area.

We also note that in lines 212-214 (previous version) the 'spatial inhomogeneities' referred to both environmental differences between ecological neighbourhoods. These arise due to difference in the total area, number of nodes, and number of edges in a given neighbourhood (connected subgraph). It is this heterogeneity that drives within metacommunity differences in local neighbourhood diversity and therefore in autonomous turnover rate. We have briefly discussed these details in the text (lines 222-225).

Line 278 correct "marcoecological." More generally, spell check the manuscript and supplement.

Corrected.

Figure S5 legend. Correct "communit".

Corrected.

Reviewer #3 (Remarks to the Author):

The authors have made substantial efforts to address most of my concerns. I agree that this work can stimulate the thinking about the internal forces of natural community turnover. The improvement on the writing is significant. A minor additional concern is the code availability for the simulation. In my opinion, for a simulation work, it is essential for others to replicate the results. Furthermore, the authors propose that autonomous turnover might be prevalent and distinguished from community turnover by external forces. For that, code availability and friendly usage will be important for more empirical ecologists to recognize this idea and apply it to future researches. The statement like “the code is available upon reasonable request” is no use because the authors can easily decline any request. I find no reason to hide the codes for a simulation study. Therefore, my suggestion for authors is providing codes with nice annotations on public repositories or appendix files for readers to replicate the shown patterns in this study.

A github code repository with a test case has been created and is now publicly available. The test case will generate the data used in Figures 5 and S3 of the manuscript. Simulations of this type are fairly CPU intensive. The examples took several hours to complete on the QMUL HPC. Hence we have also uploaded the data these simulations generate to a figshare repository. The R code included in the github repo can be used to analyse the simulation data and reproduce these key figures.

Software: https://github.com/jacobosullivan/LVMCM_src

Example data:

https://figshare.com/articles/dataset/Intrinsic_ecological_dynamics_drive_biodiversity_turnover_in_model_metacommunities_Supporting_data/14139644